# Mutant huntingtin induces neuronal apoptosis via derepressing the non-canonical poly(A) polymerase PAPD5

Zhefan Stephen Chen [1,2,16], Shaohong Isaac Peng[1,16], Lok I Leong[1], Terence Gall-Duncan[3,4], Nathan Siu Jun Wong[1], Tsz Ho Li[1], Xiao Lin [1], Yuming Wei[1], Alex Chun Koon [1], Junzhe Huang[5], Jacquelyne Ka-Li Sun[1], Clinton Turner[6], Lynette Tippett[7,8], Maurice A. Curtis [8,9], Richard L. M. Faull[7,9], Kin Ming Kwan[1,10,11], Hei-Man Chow [1,2], Ho Ko [2,5,12], Ting-Fung Chan [1,10], Kevin Talbot [13,14], Christopher E. Pearson[3,15] & Ho Yin Edwin Chan [1,2] ✉

MicroRNAs (miRNAs) are small non-coding RNAs that play crucial roles in post-transcriptional gene regulation. Poly(A) RNA polymerase D5 (PAPD5) catalyzes the addition of adenosine to the 3′ end of miRNAs. In this study, we demonstrate that the Yin Yang 1 protein, a transcriptional repressor of PAPD5, is recruited to both RNA foci and protein aggregates, resulting in an upregulation of PAPD5 expression in Huntington's disease (HD). Additionally, we identify a subset of PAPD5-regulated miRNAs with increased adenylation and reduced expression in our disease model. We focus on *miR-7-5p* and find that its reduction causes the activation of the TAB2-mediated TAK1–MKK4–JNK pro-apoptotic pathway. This pathway is also activated in induced pluripotent stem cell-derived striatal neurons and post-mortem striatal tissues isolated from HD patients. In addition, we discover that a small molecule PAPD5 inhibitor, BCH001, can mitigate cell death and neurodegeneration in our disease models. This study highlights the importance of PAPD5-mediated miRNA dysfunction in HD pathogenesis and suggests a potential therapeutic direction for the disease.

Polyglutamine (polyQ) diseases are caused by the abnormal expansion of CAG repeats in the protein-coding regions of genes involved in these diseases. These CAG repeats encode a polyQ tract that is incorporated into the affected protein[1]. Patients with polyQ diseases develop progressive neurodegeneration and motor impairment. To date, a number of polyQ disorders have been identified: Huntington's disease (HD); spinocerebellar ataxia types 1, 2, 3, 6, 7, and 17; denta-torubral pallidoluysian atrophy; and spinal and bulbar muscular

[1]School of Life Sciences, The Chinese University of Hong Kong, Hong Kong SAR, China. [2]Gerald Choa Neuroscience Institute, The Chinese University of Hong Kong, Hong Kong SAR, China. [3]Genetics & Genome Biology, The Hospital for Sick Children, Toronto, ON, Canada. [4]Molecular Genetics, University of Toronto, Toronto, ON, Canada. [5]Division of Neurology, Department of Medicine and Therapeutics, Faculty of Medicine, The Chinese University of Hong Kong, Hong Kong SAR, China. [6]Anatomical Pathology, LabPlus, Auckland City Hospital, Auckland, New Zealand. [7]School of Psychology, University of Auckland, Auckland, New Zealand. [8]University Research Centre for Brain Research, University of Auckland, Auckland, New Zealand. [9]Anatomy and Medical Imaging, University of Auckland, Auckland, New Zealand. [10]State Key Laboratory of Agrobiotechnology (CUHK), The Chinese University of Hong Kong, Hong Kong SAR, China. [11]Centre for Cell and Developmental Biology, The Chinese University of Hong Kong, Hong Kong SAR, China. [12]Li Ka Shing Institute of Health Sciences, Faculty of Medicine, The Chinese University of Hong Kong, Hong Kong SAR, China. [13]Oxford Motor Neuron Disease Centre, Nuffield Department of Clinical Neurosciences, John Radcliffe Hospital, University of Oxford, Oxford, UK. [14]Kavli Institute for Nanoscience Discovery, University of Oxford, Dorothy Crowfoot Hodgkin Building, Oxford, UK. [15]Structural Genomics Consortium, University of Toronto, Toronto, ON, Canada. [16]These authors contributed equally: Zhefan Stephen Chen, Shaohong Isaac Peng. ✉e-mail: hyechan@cuhk.edu.hk

atrophy[1]. The transcription of expanded CAG triplet repeat genomic loci leads to the production of mutant CAG transcripts, RNA foci[2], and polyQ protein aggregates[1], and subsequent neurotoxicity. The expanded CAG repeats exert pathogenic effects at the protein level, mainly through a gain-of-function effect conferred by the mutant protein containing the polyQ tract[1]. In addition to causing proteinopathy, mutant RNAs carrying the expanded CAG repeats also contribute to cellular dysfunction through a gain-of-function effect[2,3]. Notably, the transcription of expanded CAG triplet repeat sequences located in untranslated regions does not produce polyQ proteins. Li et al.[3] demonstrated that expressing such non-translatable mutant CAG repeat sequences in vivo leads to neurodegeneration.

Several studies have reported that microRNA (miRNA) homeostasis is perturbed in polyQ diseases[4], including HD[5–7]. miRNAs are small non-coding RNAs and components of the RNA-induced silencing complex (RISC), which guides the post-transcriptional silencing of mRNA targets[8]. Supplementation with miRNAs has been shown to be a rescuing strategy against HD in mice[9,10]. Understanding the regulation of miRNAs in neurons and the reasons for their dysregulation in HD can provide insights into which genes are dysregulated during disease pathogenesis and may help identify new therapeutic targets.

Poly(A) RNA polymerase D5 (PAPD5, also known as TENT4B according to nomenclature approved by the HUGO Gene Nomenclature Committee[11]) is an enzyme that promotes the addition of adenosine to the 3′ end of miRNAs[12,13]. Boele et al.[14] reported that in breast cancer cells, PAPD5 targets *miR-21* for degradation through the addition of adenosine to its 3′ end, triggering its 3′-to-5′ trimming and subsequent degradation. In this study, we found that PAPD5 is significantly upregulated in induced pluripotent stem cell (iPSC)-derived striatal neurons and post-mortem striatal tissues isolated from HD patients. We further identified a Yin Yang 1 (YY1) transcriptional regulator-binding sequence in the *PAPD5* promoter region. Mechanistically, we observed that the YY1 protein is recruited to CAG RNA foci and Htt−polyQ protein aggregates. The depletion of functional YY1 causes derepression of *PAPD5* transcription, leading to the downregulation of miRNA species. We also demonstrated that the pharmacological inhibition of PAPD5 activity rescues neuronal apoptosis and mitigates cell death in HD models. These findings indicate that PAPD5 may be a novel therapeutic target in HD.

## Results

### Expanded CAG RNA induces PAPD5 expression
Our recent RNA-seq analysis[15] showed that *PAPD5* expression is upregulated in SK-N-MC neuroblastoma cells expressing the non-translatable expanded CAG transcript $EGFP_{CAG78}$. RT-qPCR and immunoblotting analyses revealed that PAPD5 expression was induced at both the transcriptional (Fig. 1a) and translational (Fig. 1b, c) levels in $EGFP_{CAG78}$-transfected cells. This indicates that the mutant CAG transcript can induce *PAPD5* expression and that increased PAPD5 activity plays a role in RNA toxicity-mediated neurodegeneration in polyQ diseases. Prior studies, including ours, have demonstrated that expanded CAG RNA induces cell death[15–17]. In this study, when *PAPD5* expression was knocked down (Fig. 1d, e) in $EGFP_{CAG78}$-expressing SK-N-MC cells, cell death, as measured by the lactate dehydrogenase assay[15], was suppressed (Fig. 1f). This finding shows that PAPD5 plays a crucial role in CAG RNA-induced cytotoxicity. In addition to inducing in vitro cell death, the expression of untranslated CAG sequences also induces neurodegeneration in *Drosophila*[3]. When the toxic untranslated $DsRed_{CAG100}$ transgene was expressed using a *GMR-GAL4* driver, we observed an increase in *dPAPD5* expression (Fig. 1g) and photoreceptor degeneration (Fig. 1h). To determine whether the upregulation of *dPAPD5* expression plays a role in neurodegeneration, we knocked down *dPAPD5* expression in $DsRed_{CAG100}$ flies and found that retinal degeneration was rescued (Fig. 1g−i). Notably, knockdown of endogenous *PAPD5/dPAPD5* in the unexpanded CAG repeat control did

not trigger cell death in vitro (Fig. 1f) or induce neurodegeneration in vivo (Fig. 1g−i). Furthermore, the knockdown of *dPAPD5* expression did not rescue the mutant $CGG_{90}$ or $CUG_{480}$ RNA-induced external eye phenotypes in *Drosophila* (Supplementary Fig. 1). Both our in vitro and in vivo observations establish PAPD5 as a modifier of expanded CAG RNA toxicity in polyQ diseases.

### PAPD5 overexpression induces apoptosis via the TAK1−MKK4−JNK cascade
We next examined whether the overexpression of PAPD5 alone was detrimental and found that it induced caspase-3 activation (Fig. 1j, k) and cell death (Fig. 1l) in a dose-dependent manner. This induction effect was abolished (Fig. 1j−l) in cells overexpressing a catalytic-inactive PAPD5 mutant (PAPD5[D256A & D258A;18]). This highlights the role of PAPD5's catalytic function in inducing apoptotic cell death. The mitogen-activated protein kinase kinase (MKK)−c-Jun N-terminal kinase (JNK) pathway is a well-established apoptosis-induction mechanism (Fig. 1m[19]). We detected enhanced phosphorylation levels of MKK4 and JNK in PAPD5-overexpressing cells (Fig. 1n, o). Hence, we hypothesized that PAPD5 triggers apoptosis via the MKK4−JNK cascade. Transforming growth factor (TGF)-beta-activated kinase 1 (TAK1) is a MAPK kinase kinase that promotes apoptosis by activating MKK4 and JNK (Fig. 1p[20]). We observed prominent phosphorylation of TAK1 in PAPD5-overexpressing cells (Fig. 1q, r). To delineate the relationship between PAPD5 and the TAK1−MKK4−JNK-caspase cascade, we treated PAPD5-overexpressing cells with a JNK inhibitor (SP600125[21]) and found that caspase-3 cleavage was inhibited (Fig. 1s, t). SP600125 treatment also completely blocked cell death mediated by PAPD5 overexpression (Fig. 1u). Remarkably, treating PAPD5-overexpressing cells with a TAK1 kinase inhibitor (5Z-7-oxozeaenol[22]) led to dose-dependent decreases in the phosphorylation levels of TAK1 downstream effectors, including MKK4 and JNK, and in the cleavage of caspase-3 (Fig. 1v, w). Consequently, PAPD5-induced cell death was also suppressed by the TAK1 inhibitor in a dose-dependent manner (Fig. 1x). In summary, we identified a TAK1−MKK4−JNK pro-apoptotic signaling cascade mediated by PAPD5 overexpression.

We showed that expression of the expanded CAG transcript in cells induced PAPD5 expression (Fig. 1b, c) and triggered cell death (Fig. 1f). Upon *PAPD5* knockdown, mutant CAG RNA-induced cell death was rescued (Fig. 1f), accompanied by decreases in the phosphorylation levels of TAK1, MKK4, and JNK, and suppression of caspase-3 cleavage (Fig. 2a, b). Consistently, treatment with TAK1 (Fig. 2c−e) and JNK (Fig. 2f−h) inhibitors rescued mutant CAG RNA-mediated cytotoxicity. Our findings demonstrate the role of PAPD5-mediated TAK1−MKK4−JNK pro-apoptotic pathway activation in the pathogenesis of polyQ diseases.

### CAG RNA alters the cellular miRNA profile by upregulating PAPD5 expression
PAPD5 is known to decrease miRNA expression in cells[14]. We explored whether PAPD5 upregulation induced by mutant CAG RNA led to the dysregulation of cellular miRNA profiles. Among the 751 human miRNAs examined, 186 miRNAs were downregulated in mutant $EGFP_{CAG78}$-expressing cells and were restored to normal levels upon the knockdown of *PAPD5* expression (Fig. 3a, Supplementary Fig. 2, and Supplementary Data 1). This strongly suggests that PAPD5 plays a pivotal role in the dysregulation of these miRNAs in mutant CAG RNA-expressing cells. This subset of PAPD5-regulated miRNAs is thus predicted to modulate various cellular pathways, including the MAPK signaling pathway (Fig. 3b).

### CAG RNA downregulates miRNAs that target the *TAB2* gene
We showed that PAPD5 overexpression induced TAK1, MKK4, and JNK phosphorylation without changing the total protein levels of these kinases (Fig. 1n, q). By predicting the targets of the PAPD5-regulated

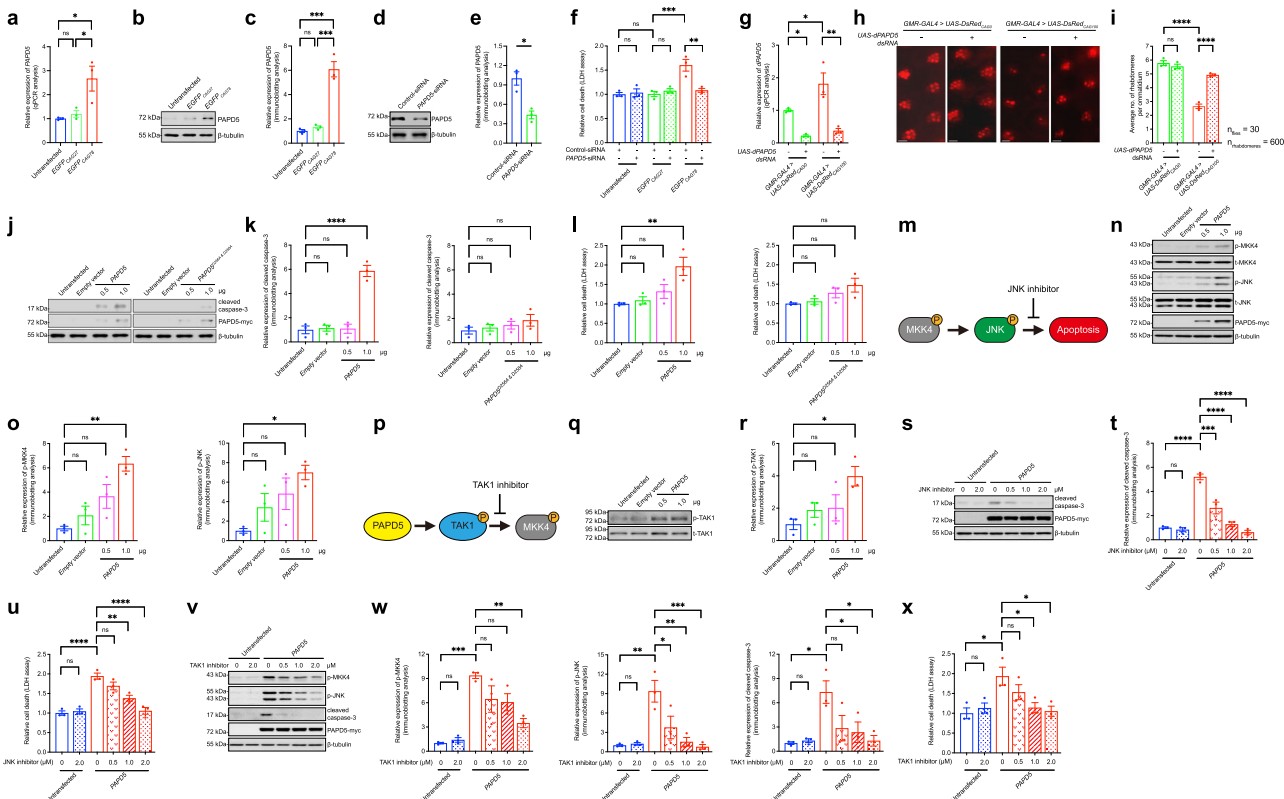

**Fig. 1 | PAPD5 upregulation activates TAK1-MKK4-JNK-caspase-3 cascade.**
**a–c** Both the transcript (**a**) and protein (**b**) levels of PAPD5 were upregulated in *EGFP_CAG78*-transfected SK-N-MC cells. **c** is the quantification of (**b**).
**d–f** Knockdown of *PAPD5* (**d**) suppressed *EGFP_CAG78*-induced cell death. Knockdown of *PAPD5* did not cause dominant cytotoxic effect in the untransfected or *EGFP_CAG27*-transfected cells. **e** is the quantification of (**d**). **g–i** Knockdown of *dPAPD5* (**g**) restored retinal degeneration in *DsRed_CAG100* flies, while it did not cause degeneration in *DsRed_CAG0* fly eyes. The genotypes of flies are listed in Supplementary Table 5. Scale bars: 10 μm. **i** is the quantification of (**h**). **j–l** Overexpression of PAPD5 is capable of inducing caspase-3 cleavage (**j**) and cell death (**l**) in a dose-dependent manner, such capability was diminished when the D256A and D258A mutations were introduced to the PAPD5 protein. **k** is the quantification of (**j**). **m** Schematic representation of the MKK-JNK apoptotic pathway. **n–o** Hyperphosphorylation of MKK4 and JNK was detected in SK-N-MC cells

overexpressed with PAPD5. **o** is the quantification of (**n**). **p** Schematic representation of the PAPD5-TAK1-MKK4 pathway. **q, r** Overexpression of PAPD5 enhanced the phosphorylation of TAK1 in a dose-dependent manner. **r** is the quantification of (**q**). JNK inhibitor (SP600125) treatment diminished the caspase-3 cleavage (**s**) and cell death (**u**) in PAPD5-transfected cells in a dose-dependent manner. **t** is the quantification of **s**. PAPD5-induced MKK4-JNK pro-apoptotic signaling pathway (**v**) and cell death (**x**) were suppressed upon treatment of TAK1 inhibitor (5Z-7-oxozeaenol) in a dose-dependent manner. **w** is the quantification of (**v**). Statistical analysis was performed using one-way ANOVA followed by *post hoc* Tukey's test, except for (**e**), in which two-tailed unpaired Student's *t*-test was used. The exact *P* values are listed in Supplementary Table 6. *n* = 3 biologically independent experiments. Data is presented as mean ± S.E.M. in (**a, c, e–g, i, k–l, o, r, t–u, w–x**). Source data are provided as a Source Data file.

miRNAs, we found that *miR-7-5p*, *miR-22-5p*, and *let-7b-5p* target TAK1 binding protein 2 (TAB2; Fig. 3c). TAB2 is crucial in modulating the activity of TAK1, its downstream MAPK kinase, and cell death pathways[23,24]. TAB2 expression increased in PAPD5-overexpressing cells (Figs. 1j, 3d, e), and this increase was proportional to the expression level of exogenous PAPD5 (Fig. 1j). In contrast, the overexpression of a catalytically inactive PAPD5 mutant (PAPD5[D256A & D258A]) did not change the TAB2 level (Figs. 1j, 3d, e).

PAPD5 promotes the reduction of miRNA by catalyzing miRNA adenylation[14]. Small RNA-seq revealed a significant increase in the adenylation level of miRNAs, including *miR-7-5p*, in PAPD5-expressing cells when compared with the untransfected control cells and cells overexpressing mutant PAPD5[D256A & D258A] (Fig. 3f). The adenylation levels of *miR-22-5p* and *let-7b-5p* were not altered (Supplementary Data 2). We further showed that *miR-7-5p* adenylation was significantly reduced when *PAPD5* was knocked down (Fig. 3g). This supports the role of PAPD5 in directly mediating the adenylation of *miR-7-5p*. Taken together, these findings indicate that the level of *miR-7-5p*, which targets *TAB2*, is reduced by excess PAPD5 protein in mutant CAG RNA-expressing cells, leading to the abnormal upregulation of TAB2 expression. Overexpression of *miR-7* could rescue cell

death in our mutant *EGFP_CAG78* RNA toxicity cell model (Fig. 3h). Western blotting analysis confirmed that the overexpression of *miR-7* suppressed *EGFP_CAG78* RNA-induced phosphorylation of TAK1 and JNK, and cleavage of caspase-3 (Fig. 3i, j). In summary, we demonstrated that a mutant CAG transcript activates the TAK1–MKK4–JNK-caspase cascade via a PAPD5-regulated *miR-7*-associated pathway (Fig. 3k). The dysregulation of miRNA has been reported in HD disease models[25] and patients[26,27]. Next, we comprehensively investigated this PAPD5-miRNA-activated apoptotic cascade in HD disease models.

## TAK1–MKK4–JNK pro-apoptotic signaling is dysregulated in HD models

When we expressed a mutant *UAS-Htt_exon1* transgene, *Htt_exon1Q93*[28], in adult *Drosophila* using the *GMR-GAL4* driver, neurodegeneration was observed in the eyes (Fig. 4a–c). *Htt_exon1Q93*-induced retinal degeneration was suppressed by knocking down *dPAPD5* expression (Fig. 4a), as shown by the restoration of rhabdomeric structures in adult fly eyes (Fig. 4b, c). Furthermore, knockdown of *dPAPD5* (Fig. 4d) mitigated the climbing defects in *Htt_exon1Q93* flies (Fig. 4e), highlighting the role of PAPD5 in modulating HD motor phenotypes.

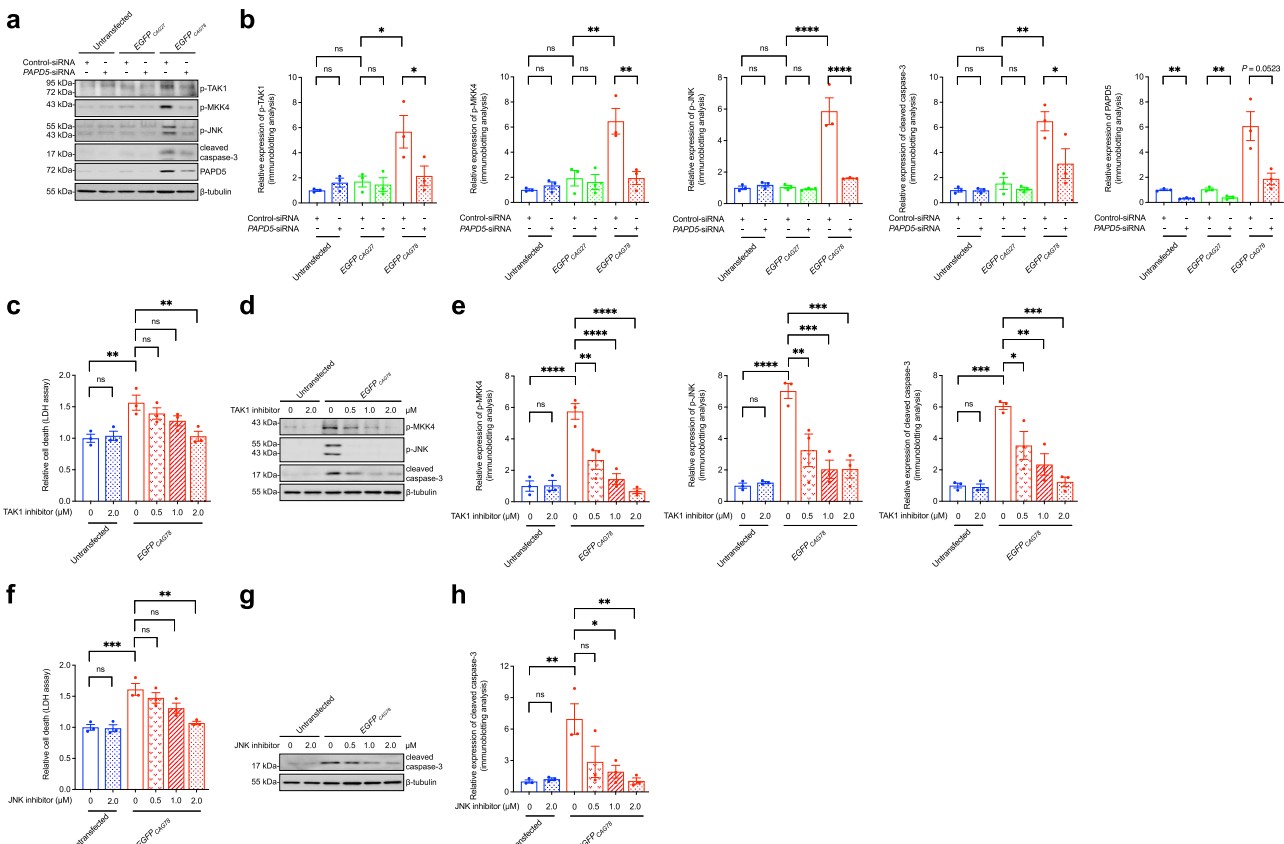

**Fig. 2 | PAPD5 activates TAK1-MKK4-JNK-caspase-3 cascade in mutant CAG RNA cells. a**, **b** Knockdown of *PAPD5* inhibited *EGFP*$_{CAG78}$-induced TAK1-MKK4-JNK phosphorylation and caspase-3 cleavage. **b** is the quantification of (**a**). **c**–**e** Mutant CAG RNA-induced cell death (**c**) and MKK4-JNK-caspase-3 activation (**d**) were rescued upon treatment of the TAK1 inhibitor (5Z-7-oxozeaenol). **e** is the quantification of **d**. **f**–**h** Treatment of JNK inhibitor (SP600125) suppressed the mutant CAG RNA-induced cell death (**f**) and caspase-3 cleavage (**g**). **h** is the quantification of (**g**). Statistical analysis was performed using one-way ANOVA followed by *post hoc* Tukey's test, except for the PAPD5 (**b**), in which two-tailed unpaired Student's *t*-test was used. The exact *P* values are listed in Supplementary Table 6. *n* = 3 biologically independent experiments. Data is presented as mean ± S.E.M. in (**b**–**c**, **e**–**f**, **h**). Source data are provided as a Source Data file.

We next expressed a mutant *Htt* construct, *EGFP-Htt*$_{1-550}$*Q89*$_{(CAG)}$, in SK-N-MC cells to confirm that PAPD5 mediates the TAK1−MKK4−JNK-caspase cascade in the HD cell model. We detected increased PAPD5 expression at both the transcriptional (Fig. 4f) and translational (Fig. 4g, h) levels in the *EGFP-Htt*$_{1-550}$*Q89*$_{(CAG)}$-transfected cells, but not in SK-N-MC cells with endogenous *Huntingtin* knocked down (Fig. 4i–k). The cytotoxicity induced by the *EGFP-Htt*$_{1-550}$*Q89*$_{(CAG)}$ construct was rescued upon the knockdown of *PAPD5* expression (Fig. 4l). The phosphorylation of TAK1, MKK4, and JNK, as well as the cleavage of caspase-3, was induced in *EGFP-Htt*$_{1-550}$*Q89*$_{(CAG)}$-expressing cells compared with unexpanded *EGFP-Htt*$_{1-550}$*Q23*$_{(CAG)}$-expressing cells and untransfected control cells (Fig. 4m, n). Upon *PAPD5* knockdown, both TAK1/MKK4/JNK phosphorylation and caspase-3 cleavage were suppressed in *EGFP-Htt*$_{1-550}$*Q89*$_{(CAG)}$-expressing cells (Fig. 4m, n). Notably, treating *EGFP-Htt*$_{1-550}$*Q89*$_{(CAG)}$-expressing cells with a JNK inhibitor suppressed caspase-3 cleavage (Fig. 4o, p) and cell death (Fig. 4q). Pharmacological inhibition of TAK1 activity in *EGFP-Htt*$_{1-550}$*Q89*$_{(CAG)}$-expressing cells terminated the activation of MKK4 and JNK and suppressed caspase-3 cleavage (Fig. 4r, s) and cell death (Fig. 4t). When the expression of TAB2 was knocked down using *miR-7*, TAK1/MKK4/JNK phosphorylation, caspase-3 cleavage (Fig. 4u, v), and cell death (Fig. 4w) were rescued in *EGFP-Htt*$_{1-550}$*Q89*$_{(CAG)}$-expressing cells. These findings suggest that PAPD5 modifies HD neurodegeneration.

## Pharmacological inhibition of PAPD5 rescues cell death and retinal degeneration in HD models

We established the pro-apoptotic properties of the PAPD5-mediated TAK1−MKK4−JNK-caspase cascade (Fig. 1j−x) and demonstrated increased PAPD5 expression in cell-derived (Fig. 4f−h) and *Drosophila*-derived (Fig. 4a) HD models. Thus, inhibiting PAPD5 activity may be an effective therapeutic strategy for HD. BCH001 is a small molecule derived from quinolone and was previously identified as a PAPD5 inhibitor by high-throughput screening[29]. We demonstrated that PAPD5 overexpression induced cell death (Fig. 1l). When PAPD5-overexpressing cells were treated with BCH001 at nanomolar concentrations, cell death (Fig. 5a), TAK1/MKK4/JNK phosphorylation, and caspase-3 cleavage (Fig. 5b, c) were suppressed in a dose-dependent manner. Consistent with the rescuing effect of miRNA overexpression (Fig. 4u, v), BCH001 treatment reduced the phosphorylation levels of TAK1, MKK4, and JNK, and suppressed caspase-3 cleavage in SK-N-MC cells transfected with the mutant *EGFP-Htt*$_{1-550}$*Q89*$_{(CAG)}$ construct (Fig. 5d, e). Cell death triggered by mutant *EGFP-Htt*$_{1-550}$*Q89*$_{(CAG)}$ expression was also rescued by BCH001 (Fig. 5f). Additionally, BCH001 treatment completely suppressed mutant *EGFP*$_{CAG78}$ RNA-induced cell death, with an IC$_{50}$ value of 16.29 nM (Supplementary Fig. 3). In addition, BCH001 suppressed retinal degeneration in *Htt*$_{exon1}$*Q93* flies (Fig. 5g, h). The moderate suppression effect may be attributable to insufficient brain uptake of BCH001 in vivo. These findings indicate that the pharmacological inhibition of PAPD5 activity offers therapeutic benefits for HD.

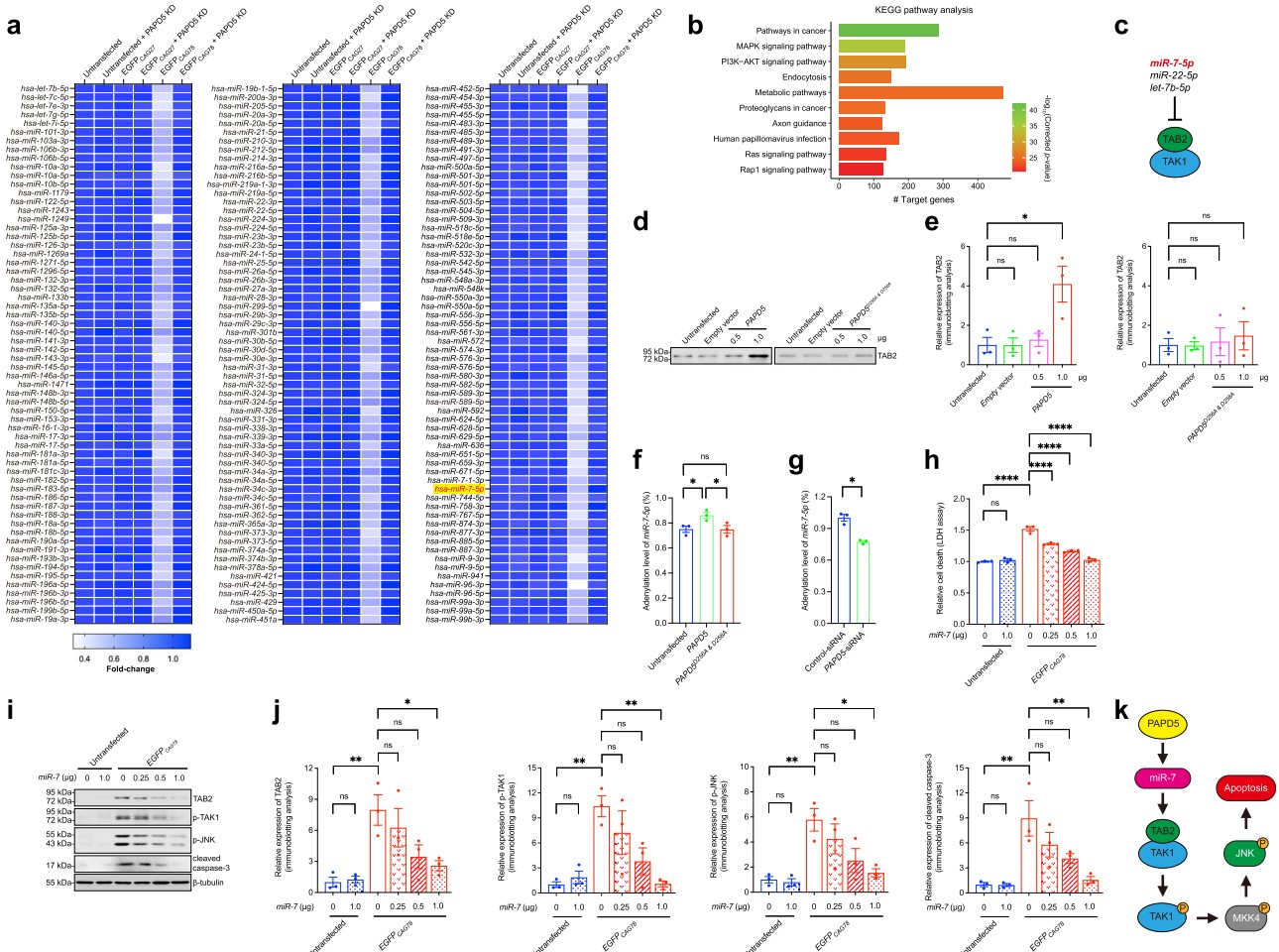

**Fig. 3 | Mutant CAG RNA induces cell death via PAPD5-manipulated miRNA species. a** Heat map analysis of the 186 miRNAs which reduction was restored upon knockdown of *PAPD5* in *EGFP_CAG78*-expressing cells. The *miR-7-5p* is highlighted. **b** Top ten enriched KEGG pathways containing the predicted mRNA targets of 186 miRNAs revealed that different cellular pathways are modulated, including the MAPK signaling pathway. **c** TAB2 is a predicted mRNA target for *miR-7-5p*, *miR-22-5p* and *let-7b-5p*. **d, e** The TAB2 protein level was upregulated in PAPD5-, but not PAPD5^D256A & D258A-overexpressing cells. **e** is the quantification of (**d**). **f** The adenylation level of *miR-7-5p* was induced in PAPD5-, but not PAPD5^D256A & D258A-overexpressing cells. **g** The adenylation level of *miR-7-5p* was reduced when *PAPD5* was

knocked down. **h–j** *EGFP_CAG78*-induced cell death (**h**) and activation of the pro-apoptotic pathway (**i**) was dose-dependently rescued upon overexpression of *miR-7*. **j** is the quantification of (**i**). **k** A schematic representation illustrates the role of *miR-7* in PAPD5-mediated TAK1-MKK4-JNK pro-apoptotic signaling pathway. Statistical analysis was performed using one-way ANOVA followed by *post hoc* Tukey's test, except for (**e, g**), in which one-way ANOVA followed by *post hoc* Fisher's LSD test and two-tailed unpaired Student's *t*-test were used, respectively. The exact *P* values are listed in Supplementary Table 6. *n* = 3 biologically independent experiments. Data is presented as mean ± S.E.M. in (**e–h, j**). Source data are provided as a Source Data file and in Supplementary Data 1 and 2.

## YY1 negatively regulates *PAPD5* transcription

We found increased transcript levels of *PAPD5* and *dPAPD5* in cell (Fig. 4f) and *Drosophila* (Fig. 4a) models of HD, indicating dysregulation of *PAPD5* transcription in HD. Next, we studied the mechanism of *PAPD5* transcriptional dysregulation. A putative YY1 transcriptional regulator-binding site (TGATGG) was identified within the *PAPD5*^+561/+860 promoter region, with this site highly conserved in mammals (Fig. 6a). YY1 is known to function as a transcriptional repressor to regulate neuronal function[30], and we have previously shown that YY1 is involved in polyQ-type spinocerebellar ataxia[31]. We constructed wild-type and mutant *PAPD5*^+561/+860 promoter reporters and performed luciferase assay to examine the role of this YY1-binding site in controlling *PAPD5* promoter activity (Fig. 6b). Our data showed that mutating the second thymine nucleotide to cytosine (T > C) in the YY1-binding site resulted in a ~2.4-fold increase in luciferase activity compared with the activity of the wild-type luciferase construct (Fig. 6b). Furthermore, *YY1* knockdown (Fig. 6c, d) led to an increase in endogenous *PAPD5* expression (Fig. 6e). We identified a putative dYY1-binding site (ATCCGCCATTT) in the *Drosophila dPAPD5* promoter (Fig. 6a) and

observed that the transcript level of *dPAPD5* was elevated (Fig. 6g) in flies upon *dYY1* knockdown (Fig. 6f). These results indicate that YY1 functions as a transcriptional repressor of *PAPD5*, and this regulatory mechanism is conserved in *Drosophila*.

## YY1 modulates PAPD5 expression and caspase-3 cleavage in mutant CAG RNA and HD cell models

Upon measuring the luciferase reporter activity of the wild-type *PAPD5*^+561/+860 promoter reporter in *EGFP_CAG78*-expressing cells, a ~2.1-fold increase in luciferase activity was detected (Fig. 6h) compared with the activity in cells expressing the unexpanded control *EGFP_CAG27*. This effect was similar to the response of the luciferase reporter activity of the mutant *PAPD5*^+561/+860 promoter reporter without *EGFP_CAG78* expression (Fig. 6b). Mutating the YY1-binding sequence abolished the induction of *PAPD5*^+561/+860 promoter activity in *EGFP_CAG78*-expressing cells (Fig. 6h). These results indicate that YY1 is involved in the transcriptional dysregulation of *PAPD5* in mutant CAG RNA-expressing cells. Subsequently, we determined whether the manipulation of YY1 affects PAPD5 expression in cells transfected

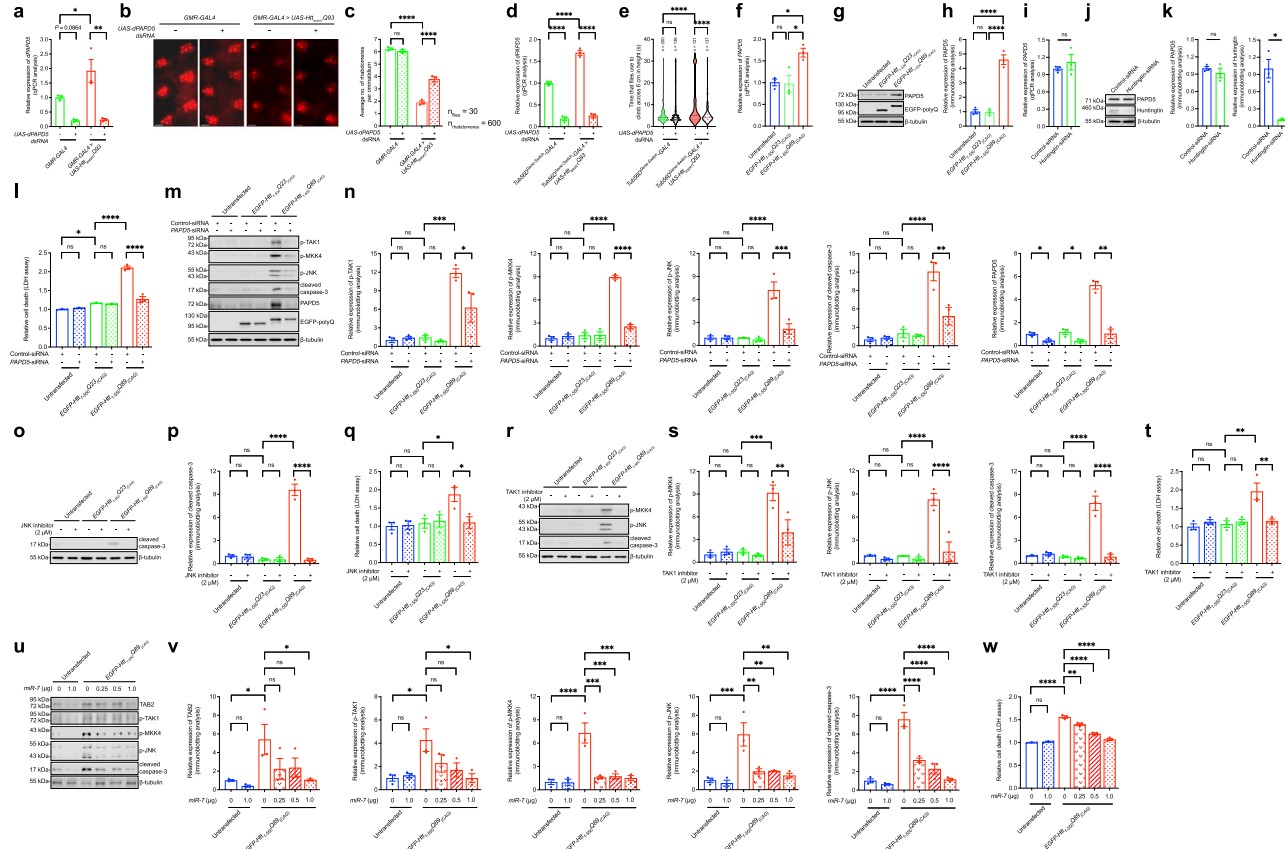

**Fig. 4 | PAPD5-mediated TAK1-MKK4-JNK pro-apoptotic signaling pathway is activated in HD. a–e** Knockdown of *dPAPD5* (**a, d**) alleviated retinal degeneration (**b**) and climbing defects (**e**) in HD flies. Knockdown of *dPAPD5* alone did not cause dominant toxicity in flies. The genotypes of flies are listed in Supplementary Table 5. Scale bars: 10 μm. **c** is the quantification of (**b**). **f–h** Both the transcript (**f**) and protein (**g**) levels of PAPD5 were upregulated in *EGFP-Htt1-550Q89(CAG)*-expressing SK-N-MC cells. **h** is the quantification of (**g**). **i–k** Knockdown of *Huntingtin* did not change the transcript (**i**) and protein (**j**) levels of PAPD5. **k** is the quantification of (**j**). **l–n** Knockdown of *PAPD5* suppressed *EGFP-Htt1-550Q89(CAG)*-induced cell death (**l**) and activation of TAK1-MKK4-JNK-caspase-3 pro-apoptotic signaling pathway (**m**). **n** is the quantification of (**m**). **o–q** JNK inhibitor (SP600125) treatment attenuated caspase-3 cleavage (**o**) and cell death (**q**) in *EGFP-Htt1-550Q89(CAG)*-

transfected SK-N-MC cells. **p** is the quantification of (**o**). **r–t** The *EGFP-Htt1-550Q89(CAG)*-induced hyperphosphorylation of MKK4 and JNK, caspase-3 cleavage (**r**) and cell death (**t**) were rescued upon treatment of TAK1 inhibitor (5Z–7-oxozeaenol). **s** is the quantification of (**r**). **u–w** Overexpression of *miR-7* suppressed the *EGFP-Htt1-550Q89(CAG)*-induced activation of TAK1-MKK4-JNK pro-apoptotic signaling pathway (**u**) and cell death (**w**). **v** is the quantification of (**u**). Statistical analysis was performed using one-way ANOVA followed by *post hoc* Tukey's test, except for (**i, k**) and the PAPD5 (**n**), in which two-tailed unpaired Student's *t*-test was used. The exact *P* values are listed in Supplementary Table 6. *n* = 3 biologically independent experiments. Data is presented as mean ± S.E.M. in (**a, c, d, f, h, i, k, l, n, p, q, s–w**). Source data are provided as a Source Data file.

---

with a mutant *EGFP_CAG78* construct. Our data showed that *YY1* knockdown increased endogenous PAPD5 expression (Fig. 6i, j), whereas the overexpression of YY1 restored PAPD5 expression in *EGFP_CAG78*-expressing cells (Fig. 6k, l). A similar effect of YY1 on PAPD5 expression was detected in cells transfected with the mutant *EGFP-Htt1-550Q89(CAG)* construct (Fig. 6m–p). Manipulating cellular YY1 levels could modulate caspase-3 cleavage in *EGFP_CAG78* and *EGFP-Htt1-550Q89(CAG)*-expressing cells (Fig. 6i–p). These findings suggest that YY1 regulates PAPD5-mediated caspase-3 activation in CAG RNA and HD cell models.

### YY1 modulates retinal degeneration in mutant CAG RNA and HD fly models

We showed that *dPAPD5* transcription was induced in the CAG RNA (Fig. 1g) and HD (Fig. 4a) fly models. Moreover, knockdown of *dYY1* (Fig. 6f) increased the *dPAPD5* transcript level (Fig. 6g). We investigated the influence of *dYY1* manipulation on retinal degeneration in mutant CAG RNA and HD flies. The knockdown of *dYY1* (Fig. 7a, d) enhanced retinal degeneration in both fly models (Fig. 7a–f), whereas *dYY1* overexpression (Fig. 7g, j) suppressed the degenerative phenotype in both mutant CAG RNA (Fig. 7g–i) and HD (Fig. 7j–l) flies. These

findings confirm that dYY1 modulates toxicity associated with CAG RNA and HD in flies.

To investigate whether dYY1 and dPAPD5 function in a linear pathway, we overexpressed *dYY1* and knocked down *dPAPD5* simultaneously in our HD fly model. Overexpression of *dYY1* and knockdown of *dPAPD5* did not yield any additive suppression effect (Fig. 7m, n), suggesting that dYY1 and dPAPD5 function in a linear pathway in HD pathogenesis.

### YY1 is recruited to CAG RNA foci and polyQ protein aggregates in HD models

We observed that the knockdown of *YY1* alone in cells upregulated *PAPD5* transcription (Fig. 6e) and stimulated caspase-3 cleavage (Fig. 6i, j). Similarly, upregulation of *dPAPD5* (Fig. 6g) and moderate retinal degeneration (Fig. 7e, f) were detected in flies with *dYY1* knockdown. We therefore hypothesized that the transcriptional dysregulation of *PAPD5/dPAPD5* and subsequent neuronal cell death resulted from a reduction in functional YY1/dYY1 protein in HD. We examined endogenous dYY1 protein expression in our HD fly model and found that its level was decreased by 3-fold (Fig. 8a). Furthermore, the dYY1 protein level remained unchanged in *DsRed_CAG100* flies (Fig. 8a). In the mammalian cell model, the YY1 protein level was consistently reduced in cells

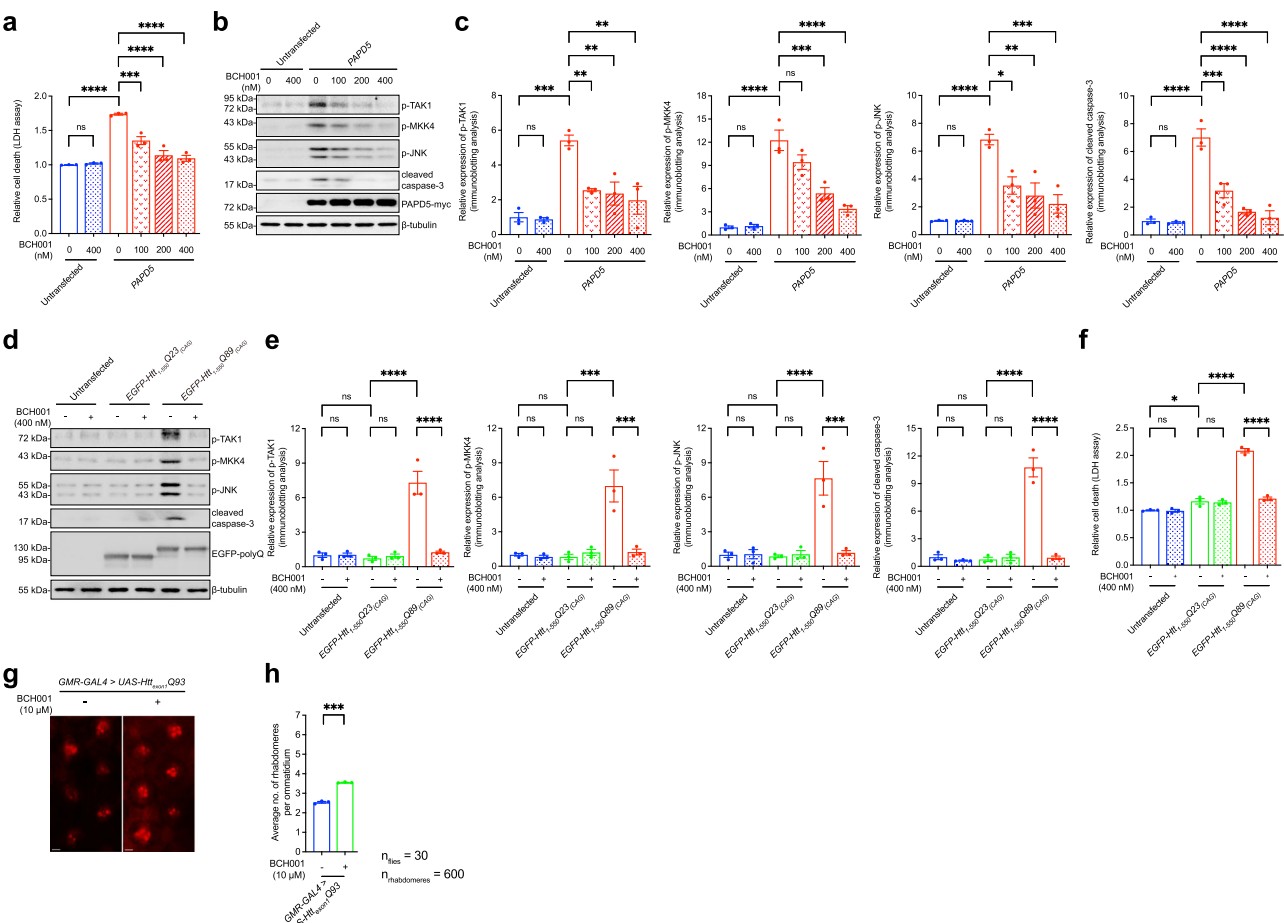

**Fig. 5 | BCH001 rescues PAPD5-activated pro-apoptotic signaling pathway in HD. a** Treatment of PAPD5 inhibitor BCH001 rescued PAPD5-induced cell death in a dose-dependent manner. **b, c** PAPD5-induced phosphorylation of TAK1, MKK4 and JNK, and caspase-3 cleavage were dose-dependently suppressed by BCH001 treatment. **c** is the quantification of (**b**). **d–f** The activation of TAK1-MKK4-JNK pro-apoptotic signaling pathway (**d**) and cell death (**f**) in *EGFP-Htt₁₋₅₅₀Q89₍CAG₎*-expressing cells were suppressed upon treatment of BCH001. **e** is the quantification of (**d**).

**g, h** BCH001 treatment ameliorated retinal degeneration in *Htt_exon1Q93* flies. Scale bars: 10 μm. **h** is the quantification of (**g**). Statistical analysis was performed using one-way ANOVA followed by *post hoc* Tukey's test, except for 5 h, in which two-tailed unpaired Student's *t*-test was used. The exact *P* values are listed in Supplementary Table 6. *n* = 3 biologically independent experiments. Data is presented as mean ± S.E.M. in (**a, c, e, f, h**). Source data are provided as a Source Data file.

transfected with the mutant *EGFP-Htt₁₋₅₅₀Q89₍CAG₎* construct (Fig. 8b–d), but it was not altered in cells expressing a non-translatable CAG RNA construct, *EGFP-STOP-Htt₁₋₅₅₀CAG91*. This construct carries tandem stop codons that terminate protein translation after the EGFP open reading frame; thus, the *Htt₁₋₅₅₀CAG91* sequence can only be transcribed but not translated (Fig. 8b). Compared with the *EGFP-Htt₁₋₅₅₀Q89₍CAG₎* construct, which produced both RNA foci and polyQ protein aggregates, we detected the formation of only RNA foci but not polyQ protein aggregates in cells expressing *EGFP-STOP-Htt₁₋₅₅₀CAG91* (Supplementary Fig. 4a). We generated another construct, *EGFP-Htt₁₋₅₅₀Q89₍CAA/G₎*, in which the polyQ tract was encoded by interrupted CAA/CAG codons to disrupt the continuity of the CAG repeat sequence[3]. This construct did not produce CAG RNA foci but facilitated the production of polyQ protein aggregates (Fig. 8b and Supplementary Fig. 4a). Thus, we predicted that the polyQ protein aggregate produced by the *EGFP-Htt₁₋₅₅₀Q89₍CAA/G₎* construct would interfere with the endogenous YY1 level. As expected, we detected a 3-fold reduction in the YY1 protein level in *EGFP-Htt₁₋₅₅₀Q89₍CAA/G₎*-expressing cells (Fig. 8b–d). Our results indicate that Htt–polyQ protein aggregates, rather than CAG RNA foci, play a critical role in reducing the YY1 protein level in HD disease models.

Interestingly, although the YY1 protein level was not altered in cells transfected with *EGFP-STOP-Htt₁₋₅₅₀CAG91*, PAPD5 induction could still be detected (Fig. 8b–d and Supplementary Fig. 4a). This

suggests that YY1's function was compromised by mutant CAG RNA expression. In polyQ diseases, both mutant CAG RNA foci and polyQ protein aggregates possess cellular protein-sequestration capabilities[17,32,33]. Using fluorescence in situ hybridization (FISH) coupled with immunocytochemistry (ICC), we observed that endogenous YY1 protein (in green) was recruited to both CAG RNA foci and polyQ protein aggregates in *EGFP-Htt₁₋₅₅₀Q89₍CAG₎*-transfected cells (Fig. 8e–g). In *EGFP-STOP-Htt₁₋₅₅₀CAG91*- and *EGFP-Htt₁₋₅₅₀Q89₍CAA/G₎*-expressing cells, YY1 co-localized with CAG RNA foci (Fig. 8e, f) and polyQ protein aggregates (Fig. 8g), respectively. Chromatin immunoprecipitation sequencing (ChIP-seq) was performed to investigate whether the function of YY1 was perturbed in our HD cell model. Compared with the unexpanded control cells, YY1's binding to its DNA targets, including *PAPD5* (Fig. 8h, i), was reduced in *EGFP-Htt₁₋₅₅₀Q89₍CAG₎*-expressing cells (Supplementary Data 3). Our findings support the notion that functional YY1 protein is depleted in mutant Htt–polyQ models through its recruitment to both RNA foci and protein aggregates, causing an increase in PAPD5 expression.

## Both CAG RNA foci and polyQ protein aggregates contribute to PAPD5-mediated apoptotic signaling and cell death

The Htt constructs (Fig. 8b) were used to examine the activity of PAPD5-mediated pro-apoptotic signaling and cell death. The most

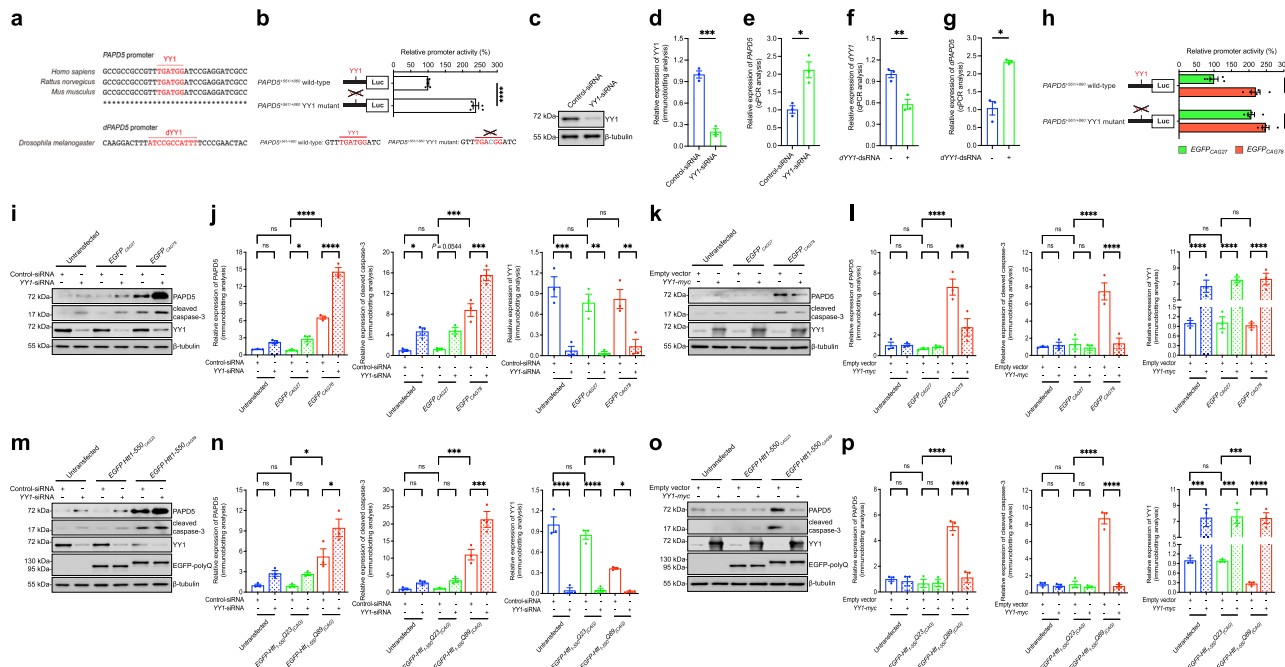

**Fig. 6 | YY1 modulates PAPD5 level and cytotoxicity in cell CAG RNA and HD models. a** The putative YY1 binding site (TGATGG) within the *PAPD5^{+561/+860}* promoter region is conserved in mammals. A putative dYY1 binding site (ATCCGC-CATTT) was identified in *dPAPD5* promoter region. **b** Luciferase reporter analysis showed that a single point mutation in the YY1 binding site within the *PAPD5^{+561/+860}* promoter region increased luciferase activity. **c–g** Knockdown of *YY1/dYY1* elevated *PAPD5/dPAPD5* transcript level. **d** is the quantification of (**c**). **h** Luciferase activity of wild-type *PAPD5^{+561/+860}* promoter reporter was induced in *EGFP_{CAG78}*-expressing cells, while such induction was abolished when the YY1 binding site was mutated. **i–l** Knockdown of *YY1* increased the PAPD5 and cleaved caspase-3 protein levels (**i**) in *EGFP_{CAG78}*-expressing cells, whereas overexpression of YY1 exerted the opposite effect (**k**). **j, l** are the quantifications of (**i, k**), respectively. **m–p** Knockdown of *YY1* increased PAPD5 and cleaved caspase-3 protein levels (**m**) in *EGFP-Htt_{1-550}Q89_{(CAG)}*-expressing cells, whereas overexpression of YY1 exerted the opposite effect (**o**). **n, p** are the quantifications of (**m, o**), respectively. Statistical analysis was performed using one-way ANOVA followed by *post hoc* Tukey's test, except for (**d–g**), in which two-tailed unpaired Student's *t*-test was used. The exact *P* values are listed in Supplementary Table 6. *n* = 3 biologically independent experiments. Data is presented as mean ± S.E.M. in (**d–g, j, l, n, p**). Source data are provided as a Source Data file.

prominent induction of TAK1–MKK4–JNK signaling (Fig. 8j, k and Supplementary Fig. 4b) and cell death (Fig. 8l) was detected in *EGFP-Htt_{1-550}Q89_{(CAG)}*-expressing SK-N-MC cells. This induction, although at a lower level, was also detected in cells transfected with *EGFP-STOP-Htt_{1-550}CAG91* or *EGFP-Htt_{1-550}Q89_{(CAA/G)}* (Fig. 8j–l and Supplementary Fig. 4b). These results highlight the contributions of both CAG RNA foci and polyQ protein aggregates to the activation of PAPD5-mediated pro-apoptotic signaling in HD cell models.

### Role of the YY1-PAPD5 apoptotic pathway in HD patient iPSCs-derived striatal neurons and post-mortem brain samples

To further explore the activation of the YY1-PAPD5 pro-apoptotic signaling pathway in HD patient neurons, we differentiated HD patient iPSCs into striatal neurons (Supplementary Fig. 5a–c[34]). FISH and ICC experiments revealed that endogenous YY1 protein co-localized with both CAG RNA foci (Fig. 9a) and Htt-polyQ protein aggregates (Fig. 9b) in diseased neurons. Notably, the PAPD5 protein level was significantly increased in HD striatal neurons compared with the controls (Fig. 9c, d). Phosphorylation of MKK4 and JNK, as well as cleavage of caspase-3, were induced in HD patient striatal neurons, but this induction was suppressed upon treatment with BCH001 (Fig. 9e, f and Supplementary Fig. 5d). These findings emphasize the activation of a PAPD5-mediated apoptotic pathway in HD striatal neurons and suggest that PAPD5 inhibition could be a potential therapeutic strategy against HD toxicity.

We also detected increased PAPD5 expression at both the transcriptional (Fig. 9g) and translational (Fig. 9h, i) levels in the post-mortem striatal tissues of HD patients (Supplementary Table 3[35]). Furthermore, RT-qPCR showed a reduced expression of *miR-7-5p* (Fig. 9j) and immunoblotting confirmed an increase in TAB2 (Fig. 9k, l) level in striatal tissues from patients. We further examined cerebellar

tissues from the same set of HD patients and unaffected individuals to determine whether the PAPD5-mediated pathway was selectively activated in the striatum. Notably, the increase in PAPD5 protein levels was more pronounced in HD striatum (4.5-fold; Fig. 9h, i) than in the cerebellum (2-fold; Fig. 9n, o). Although the *PAPD5* transcript and protein levels were significantly upregulated in the HD patient cerebellar tissues (Fig. 9m–o), no significant change was observed in *miR-7-5p* and TAB2 levels (Fig. 9p–r). This suggests that PAPD5 induction beyond a certain threshold may trigger miRNA dysregulation and the downstream apoptotic pathway. Importantly, the striatum is particularly vulnerable to neurodegeneration in HD patients, while the cerebellum is less affected[36]. This suggests that the selective activation of PAPD5-mediated apoptotic pathway mirrors tissue-specific patterns of neurodegeneration in HD.

## Discussion

miRNAs are small regulatory RNAs that are widely expressed in tissues, including the brain. They control gene expression by silencing their mRNA targets via the RNA interference mechanism mediated by RISC[4]. miRNA dysregulation has been identified in the affected brain regions of patients with HD[26], indicating that alterations in the mRNA levels of miRNA targets may contribute to disease pathogenesis. In this study, we demonstrated that the TAK1–MKK4–JNK pro-apoptotic pathway is activated in HD models owing to the downregulation of miRNAs, including *miR-7-5p* (Fig. 10).

### Non-canonical poly(A) polymerase PAPD5 is involved in miRNA dysregulation in HD

PAPD5 is a non-canonical poly(A) polymerase that catalyzes the post-transcriptional adenylation of miRNAs[12,13]. This regulatory mechanism

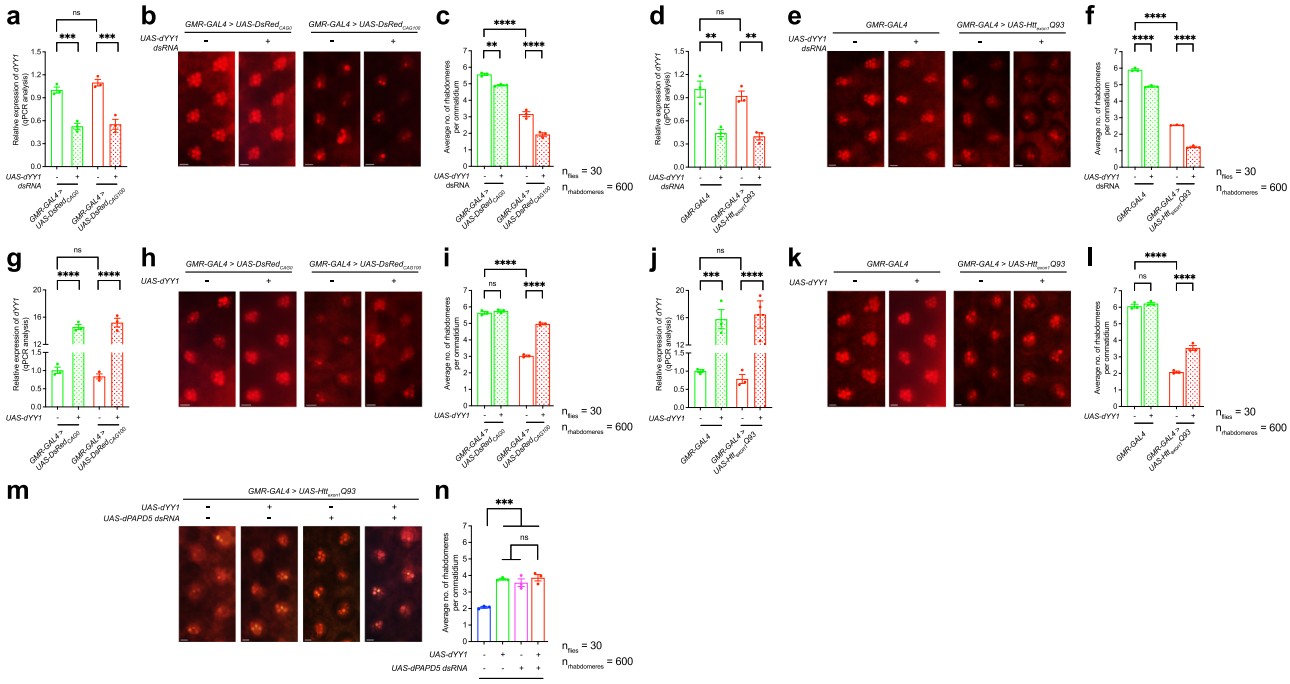

**Fig. 7 | YY1 modulates cytotoxicity in fly CAG RNA and HD models.**
**a**–**c** Knockdown of *dYY1* (**a**) intensified retinal degeneration in *DsRed$_{CAG100}$* flies. Knockdown of *dYY1* also caused moderate degeneration in *DsRed$_{CAG0}$* fly eyes. **c** is the quantification of (**b**). **d**–**f** Knockdown of *dYY1* (**d**) intensified retinal degeneration in *Htt$_{exon1}$Q93* flies. Knockdown of *dYY1* alone also caused moderate degeneration in fly eyes. **f** is the quantification of (**e**). **g**–**i** Overexpression of *dYY1* (**g**) restored retinal degeneration in *DsRed$_{CAG100}$* flies, while it did not cause degeneration in *DsRed$_{CAG0}$* fly eyes. **i** is the quantification of (**h**). **j**–**l** Overexpression of *dYY1* (**j**) restored retinal degeneration in *Htt$_{exon1}$Q93* flies. dYY1 did not cause

degeneration in fly eyes when overexpressed alone. **l** is the quantification of (**k**). **m**, **n** Overexpression of *dYY1* and knockdown of *dPAPD5* simultaneously did not generate additive suppression effect when compared to the *dYY1* overexpression or *dPAPD5* knockdown alone. **n** is the quantification of (**m**). The genotypes of flies are listed in Supplementary Table 5. Scale bars: 10 μm. Statistical analysis was performed using one-way ANOVA followed by *post hoc* Tukey's test. The exact *P* values are listed in Supplementary Table 6. *n* = 3 biologically independent experiments. Data is presented as mean ± S.E.M. in (**a**, **c**, **d**, **f**, **g**, **i**, **j**, **l**, **n**). Source data are provided as a Source Data file.

has been reported in cancers[14,37]. Boele et al.[14] reported a positive correlation between PAPD5 expression and the adenylation level of *miR-21* across multiple types of cancer. Mechanistically, PAPD5 mediates the addition of adenosine to the 3′ end of *miR-21*, which promotes its 3′-to-5′ trimming and degradation[14]. Our study further demonstrated the role of PAPD5-catalyzed miRNA adenylation in HD pathogenesis. We showed that the PAPD5 transcript and protein levels were increased in HD models (Fig. 4a, f–h), including patient-derived striatal neurons (Fig. 9c, d) and post-mortem striatal tissues (Fig. 9g–i). To determine the functional consequences of PAPD5 upregulation, we identified 186 miRNAs with reduced levels in our mutant CAG RNA-expressing cell model (Supplementary Data 1). Upon knocking down *PAPD5* expression, the expression of these miRNAs was restored to control levels (Supplementary Data 1). Small RNA-seq revealed that out of the 186 miRNAs, the adenylation levels of 11 miRNAs, namely *miR-7-5p* (Fig. 3f), *let-7c-5p* (Supplementary Fig. 6a), *let-7g-5p* (Supplementary Fig. 6b), *let-7i-5p* (Supplementary Fig. 6c), *miR-10a-5p* (Supplementary Fig. 6d), *miR-101-3p* (Supplementary Fig. 6e), *miR-196a-5p* (Supplementary Fig. 6f), *miR-28-3p* (Supplementary Fig. 6g), *miR-30d-5p* (Supplementary Fig. 6h), *miR-424-5p* (Supplementary Fig. 6i), and *miR-589-5p* (Supplementary Fig. 6j), were significantly elevated by the wild-type PAPD5 protein, but not by its catalytic-inactive mutant (PAPD5$^{D256A \& D258A}$). It is noteworthy that the levels of adenylated miRNAs were not substantially increased in wild-type PAPD5-expressing cells (Fig. 3f and Supplementary Fig. 6), this could be attributable to the rapid degradation of adenylated miRNAs[14]. Taken together, these findings suggest that the reduction of this subset of 11 miRNAs is directly mediated by PAPD5-catalyzed adenylation in mutant CAG RNA-expressing cells. The remaining miRNAs that were downregulated in mutant CAG RNA cells and restored upon knockdown of *PAPD5*

could be regulated through alternative mechanisms. Several PAPD5-adenylated miRNAs have been shown to influence the expression of chromatin modifiers, which, in turn, affect miRNA levels[38–46]. For example, *miR-589-5p* and *miR-101-3p* have been reported to inhibit the expression of *histone deacetylase 3* (*HDAC3*)[46] and *histone deacetylase 9* (*HDAC9*)[40], respectively. Both HDAC3 and HDAC9 have been implicated in repressing the expression of various miRNAs, such as the miR-17 ~ 92 family of miRNA clusters[47,48]. The miR-17 ~ 92 cluster members, including *miR-17-5p*, *miR-18a-5p*, *miR-19b-1-5p*, and *miR-20a-5p*, were found to be downregulated in our mutant CAG RNA cell model and whose expression could be restored following *PAPD5* knockdown (Fig. 3a). In summary, the widespread downregulation of miRNAs can be directly triggered by PAPD5-mediated adenylation and indirectly influenced through secondary pathways activated by PAPD5.

## YY1 dysfunction causes PAPD5 induction in HD

We clarified the role of PAPD5 in RNA- and protein-associated neurodegeneration and provided a mechanistic explanation for the transcriptional upregulation of PAPD5 in HD. We showed that both protein aggregates and RNA foci contributed to PAPD5 dysregulation in our HD cell model (Fig. 8c). Mutant polyQ protein aggregates are known to sequester essential cellular proteins and remove them from their normal subcellular locations, leading to neuronal dysfunction[1]. For example, the CREB-binding protein is a well-characterized transcriptional regulator with compromised activity in polyQ diseases owing to its recruitment to protein aggregates[32]. YY1 functions as a transcriptional regulator in the mammalian nervous system to regulate gene expression[30]. We previously reported that YY1 is recruited to Ataxin-3–polyQ protein aggregates, resulting in transcriptional dysregulation that contributes to SCA3 neurodegeneration[31]. In addition to protein

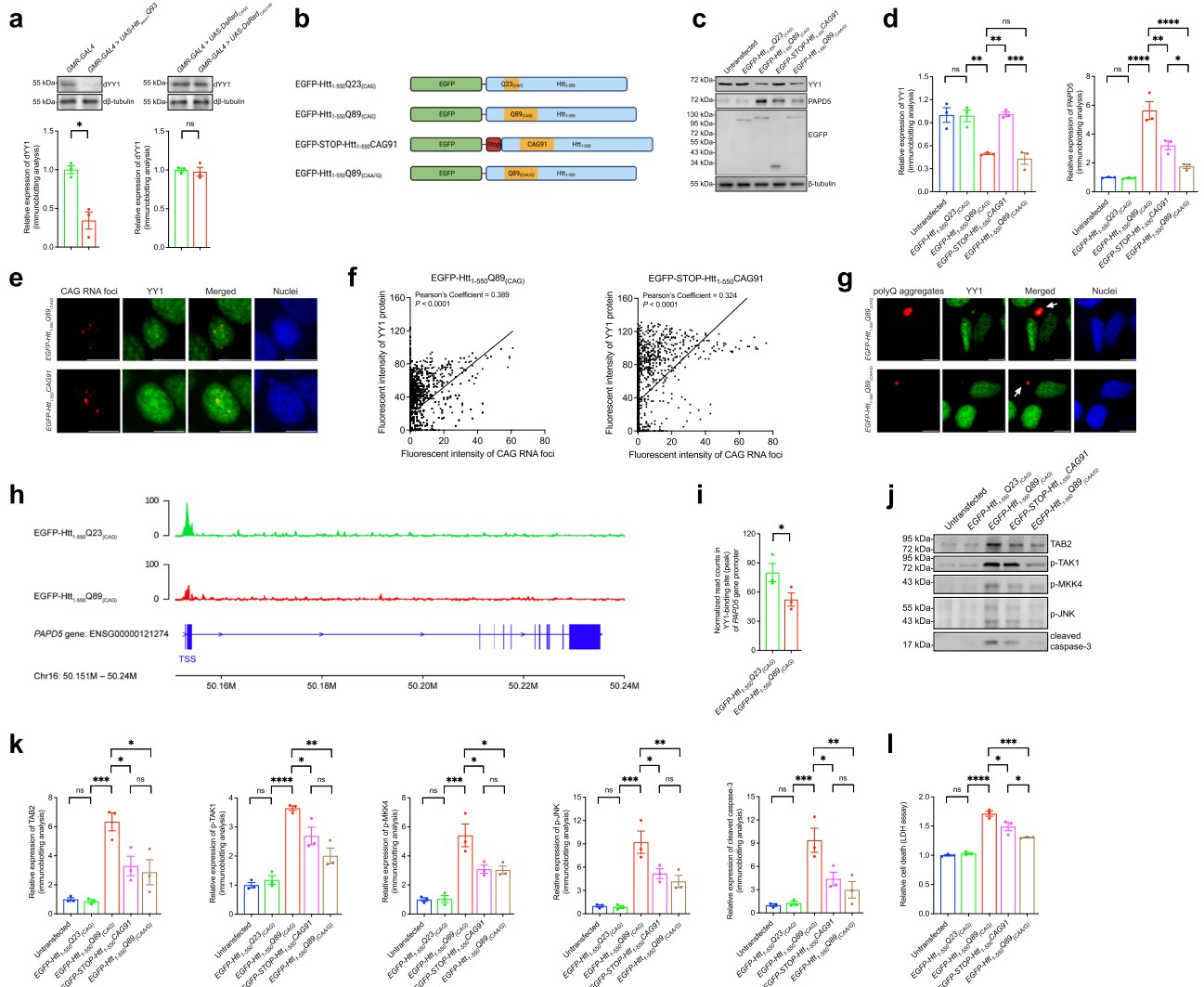

**Fig. 8 | The recruitment of YY1 protein to CAG RNA foci and polyQ protein aggregates leads to PAPD5 induction and activation of TAK1-MKK-JNK pro-apoptotic signaling. a** The dYY1 protein level was reduced in HD, but not mutant CAG RNA fly head samples. **b** Schematic representation of the different Htt constructs used in this study. Created in BioRender. Chen, S. (2025) https://BioRender. com/v18l880. **c, d** Soluble YY1 protein level was reduced in *EGFP-Htt₁-₅₅₀Q89(CAG)*- and *EGFP-Htt₁-₅₅₀Q89(CAA/G)*-transfected cells. The highest induction level of PAPD5 expression was detected in *EGFP-Htt₁-₅₅₀Q89(CAG)*-expressing cells, while *EGFP-STOP-Htt₁-₅₅₀CAG91* was more capable than *EGFP-Htt₁-₅₅₀Q89(CAA/G)* in terms of inducing PAPD5 level. **d** is the quantification of (**c**). **e** The FISH experiments showed the presence of CAG RNA foci in *EGFP-Htt₁-₅₅₀Q89(CAG)*- and *EGFP-STOP-Htt₁-₅₅₀CAG91*-expressing cells. YY1 protein (green) was found co-localized with CAG RNA foci (red). Cell nuclei (blue) were stained with Hoechst 33342. Scale bars: 10 μm. **f** The co-localization between YY1 and CAG RNA foci was statistically significant. **g** YY1 (green) was recruited (arrows) to polyQ protein aggregates (red) in *EGFP-Htt₁-₅₅₀Q89(CAG)*- and *EGFP-Htt₁-₅₅₀Q89(CAA/G)*-expressing cells. Cell nuclei (blue)

were stained with Hoechst 33342. Scale bars: 10 μm. **h** Representative plot to demonstrate the YY1-binding sites (peaks) at *PAPD5* gene promoter region. The binding of YY1 to *PAPD5* promoter was reduced in *EGFP-Htt₁-₅₅₀Q89(CAG)*-expressing cells. TSS denotes the transcription start site. **i** The normalized read counts in YY1-binding site (peak) of *PAPD5* gene promoter was reduced in *EGFP-Htt₁-₅₅₀Q89(CAG)*-expressing cells. **j–l** The most prominent induction of TAK1-MKK4-JNK pro-apoptotic signaling (**j**) and neuronal cell death (**l**) was detected in *EGFP-Htt₁-₅₅₀Q89(CAG)*-expressing cells. When compared to *EGFP-Htt₁-₅₅₀Q89(CAA/G)*, *EGFP-STOP-Htt₁-₅₅₀CAG91* was more capable of stimulating the TAK1-MKK4-JNK pro-apoptotic signaling and neuronal cell death. **k** is the quantification of (**j**). Statistical analysis was performed using one-way ANOVA followed by *post hoc* Tukey's test, except for 8a and 8i, in which two-tailed unpaired Student's *t*-test and one-tailed unpaired Student's *t*-test were used, respectively. The exact *P* values are listed in Supplementary Table 6 and Supplementary Data 3. *n* = 3 biologically independent experiments. Data is presented as mean ± S.E.M. in (**a, d, i, k, l**). Source data are provided as a Source Data file and in Supplementary Data 3.

aggregates, repeat expansion RNAs also possess protein-sequestration capabilities[49]. In this study, we showed that YY1 is recruited to both CAG RNA foci and Htt–polyQ protein aggregates (Fig. 8e–g). Our ChIP-seq analysis revealed a significant reduction of YY1's binding to *PAPD5* gene promoter (Fig. 8h, i). Although this reduction was modest (-1.5-fold), we observed a significant induction of PAPD5 (Fig. 8c, d), which led to apoptotic induction in our HD cell model (Fig. 8j, k). Notably, the functional depletion of YY1 markedly decreases the enrichment of epigenetic modifiers at gene promoter regions, thereby enhancing

transcriptional activation[50]. Therefore, even a moderate disruption of YY1 binding at the *PAPD5* promoter can be sufficient to lift epigenetic repression and trigger *PAPD5* expression. In addition to *PAPD5*, YY1's binding to other targets[51], including Hippo pathway components and WNT signaling molecules, was perturbed in our HD cell model (Supplementary Data 3). Notably, Hippo pathway and WNT signaling are both implicated in HD[52]. This suggests additional YY1 dysfunction-triggered pathogenic pathways, further highlighting the crucial role of YY1 in HD pathogenesis.

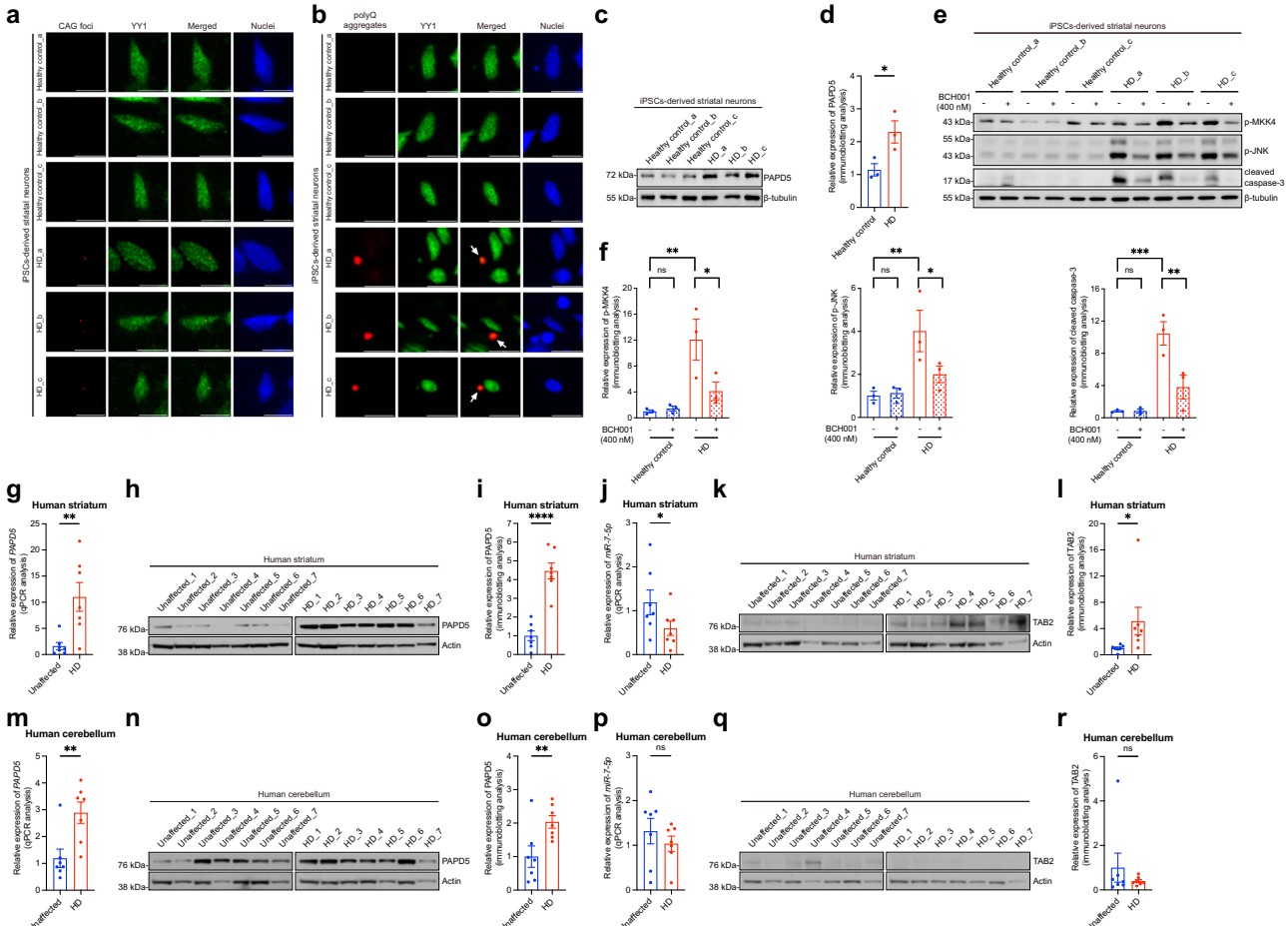

**Fig. 9 | The YY1-PAPD5 pro-apoptotic signaling pathway is activated in HD patient neurons.** Immunofluorescent analysis showed that both CAG RNA foci (**a**) and polyQ protein aggregates (**b**) were detected in HD iPSCs-derived striatal neurons. YY1 protein was found co-localized with CAG RNA foci and recruited (arrows) to polyQ protein aggregates. Cell nuclei (blue) were stained with Hoechst 33342. Scale bars: 10 μm. The experiment was repeated independently for 3 times with similar results, and representative images are shown. **c, d** The PAPD5 protein level was upregulated in HD patient striatal neurons when compared to the healthy control striatal neurons. **d** is the quantification of (**c**). **e, f** Treatment of BCH001 suppressed the activation of the PAPD5-mediated apoptotic pathway in HD patient iPSCs-derived striatal neurons. **f** is the quantification of (**e**). **g–i** The *PAPD5* transcript (**g**) and protein (**h**) levels were significantly induced in HD patient striatal samples. **i** is the quantification of (**h**). **j** The level of *miR-7-5p* was significantly

reduced in HD patient striatal samples. **k, l** The protein level of TAB2 was significantly upregulated in the striatal tissues from HD patients. **l** is the quantification of **k**. **m–o** The transcript (**m**) and protein (**n**) levels of PAPD5 were significantly upregulated in the HD cerebellar tissues. **o** is the quantification of (**n**). **p–r** No alteration of *miR-7-5p* (**p**) and TAB2 (**q**) levels was detected in the HD cerebellar samples. **r** is the quantification of (**q**). Statistical analysis was performed using one-tailed unpaired Student's *t*-test, except for (**f**), in which one-way ANOVA followed by *post hoc* Fisher's LSD test was used. The exact *P* values are listed in Supplementary Table 6. Each data point in Fig. (**d**, **f**) represents the average value obtained from three independent differentiation for each of the healthy control and HD iPSCs. Data is presented as mean ± S.E.M. in (**d**, **f**, **g**, **i**, **j**, **l**, **m**, **o**, **p**, **r**). Source data are provided as a Source Data file.

## CAG RNA foci and polyQ protein aggregates cause YY1 dysfunction and trigger cell death in HD

YY1 is a multidomain protein[53] that carries an internal REcruitment of POlycomb (REPO) domain, which is essential for its recruitment to polyQ protein aggregates[31]. In addition, YY1 carries a C-terminal zinc-finger (ZF) domain that is required for interacting with RNAs[54,55]. The recruitment of YY1 to Htt–polyQ protein aggregates and CAG RNA foci may thus be attributable to the presence of REPO and ZF domains, respectively. We showed that both polyQ protein aggregates and CAG RNA foci induce PAPD5-mediated pro-apoptotic signaling. Interestingly, under similar transfection conditions (Supplementary Fig. 4c), compared with *EGFP-Htt$_{1-550}$Q89$_{(CAA/G)}$*, *EGFP-STOP-Htt$_{1-550}$CAG91* is more capable of stimulating the TAK1–MKK4–JNK pathway (Fig. 8j, k) and cell death (Fig. 8l). Consistently, PAPD5 expression is induced to a higher level in *EGFP-STOP-Htt$_{1-550}$CAG91*-expressing cells than in *EGFP-Htt$_{1-550}$Q89$_{(CAA/G)}$*-expressing cells (Fig. 8c, d). This implies that mutant CAG RNA contributes more to the activation of PAPD5-mediated

apoptotic signaling and cell death than aggregated polyQ protein in our HD cell model.

## HD pathogenesis involves the TAK1–MKK4–JNK pro-apoptotic signaling pathway

TAK1 signaling is associated with apoptosis[24,56], and its inhibition has been shown to provide neuroprotection in traumatic brain injury[57]. The kinase activity of TAK1 is regulated by TABs[58], with TAB2 being one of the known activators[23]. In this study, we found that PAPD5 and TAB2 were concomitantly upregulated in the HD cell model (Fig. 4g, h, u, v) and patient post-mortem striatal tissues (Fig. 9h, i, k, l). We identified three PAPD5-regulated miRNAs that target *TAB2* (Fig. 3c), and the *TAB2* transcript level was reported to be increased in the caudate nucleus of HD patient brains[59]. The dysregulation of TAB2 expression is expected to affect TAK1 phosphorylation and trigger the downstream pro-apoptotic pathway in HD. In addition to TAB2-targeted miRNAs, *miR-143-3p*, which directly targets *TAK1*[60,61], is downregulated in our mutant

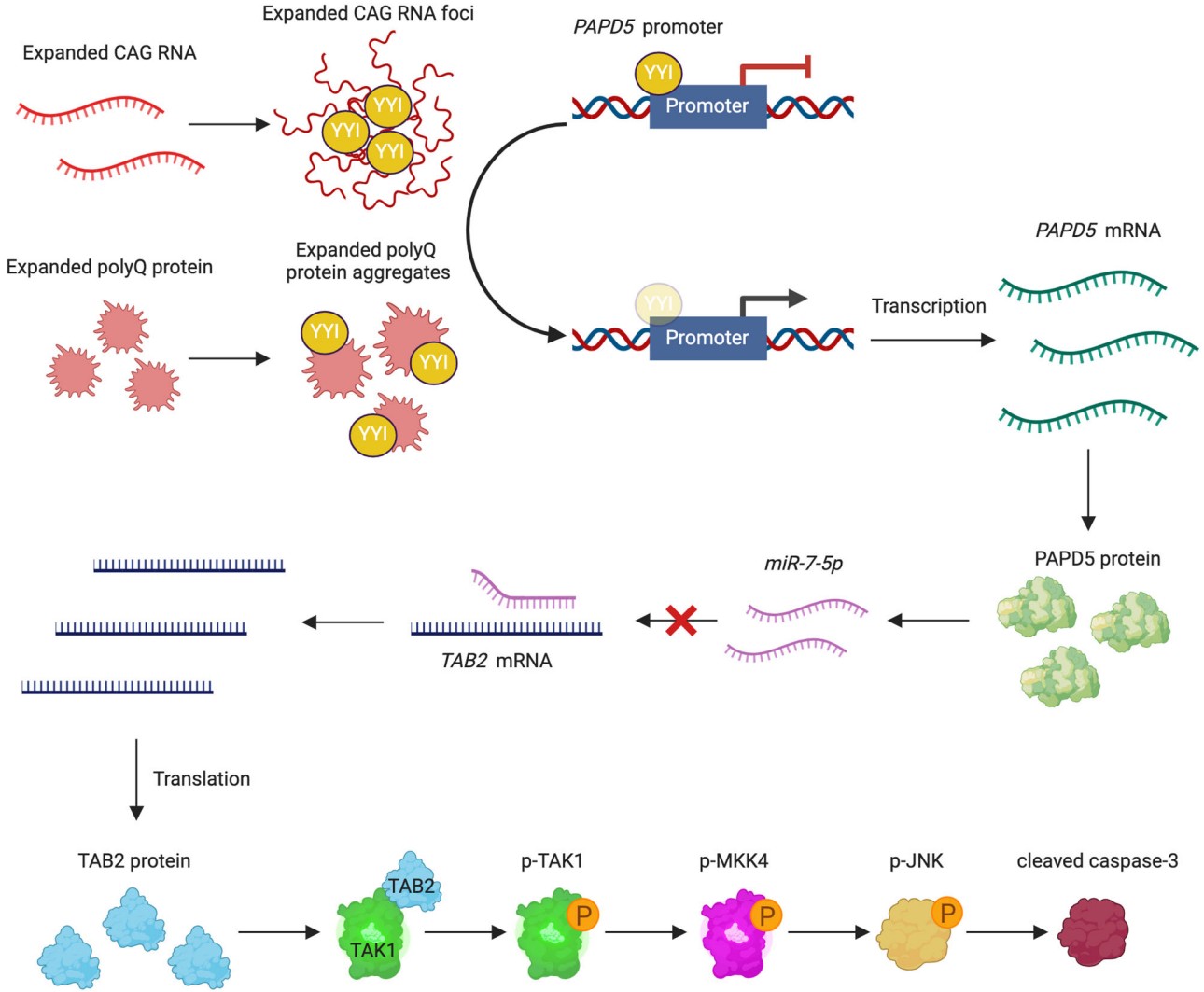

**Fig. 10 | A proposed model for PAPD5-mediated TAK1-MKK4-JNK pro-apoptotic signaling pathway in HD.** In HD, YY1 protein is recruited by CAG RNA foci and polyQ protein aggregates and this results in the derepression of *PAPD5* transcription. Upregulation of PAPD5 expression leads to the induction of TAB2, which triggers TAK1-MKK4-JNK-caspase cascade, and eventually results in caspase-3 activation and neuronal cell death. Created in BioRender. Chen, S. (2025) https://BioRender.com/d62n794.

CAG RNA cell model and restored upon *PAPD5* knockdown (Supplementary Data 1). It has been reported that *miR-143-3p* downregulation promotes TAK1 phosphorylation in the liver[61]. In ovarian cancer, *miR-143-3p* downregulates TAK protein levels[60]. The mechanism underlying the role of *miR-143-3p* in HD remains to be elucidated. The TGF-beta receptor associates with TAK1 and TNF receptor-associated factor 6 (TRAF6). The ligand-bound TGF-beta receptor induces TRAF6 K-63 poly-ubiquitination, which activates TAK1 function[62]. TRAF6 protein has been found to accumulate in the insoluble fraction of post-mortem brains of HD patients[63], suggesting its potential role in regulating TAK1 and contributing to the activation of TAK1 in HD.

### BCH001 is a modulator of the PAPD5-mediated pro-apoptotic pathway and a potential drug candidate against HD toxicity

PAPD5 is a non-canonical poly(A) polymerase with enzymatic activity, making it a potential drug target[14]. A small molecule, BCH001, was recently reported to specifically inhibit PAPD5 enzymatic activity[29]. Upon treatment with BCH001, the 3' adenylation of the non-coding RNA template telomerase RNA component was reduced[29]. This compound did not induce cell death at 400 nM in our cell model (Fig. 5a), and this dose was sufficient to suppress the PAPD5-mediated

activation of TAK1–MKK4–JNK pro-apoptotic signaling (Fig. 5b, c) and cell death (Fig. 5a). Furthermore, BCH001 treatment ameliorated cell death (Fig. 5f) and retinal degeneration (Fig. 5g, h) in cell and fly HD models, suggesting its potential as an inhibitor of PAPD5-mediated apoptotic cell death. Our findings highlight a new therapeutic direction for HD.

## Methods
### Ethics statement
Biological and chemical safety approval (14122815) from The Chinese University of Hong Kong was obtained for this study. All animal procedures were approved by the CUHK Animal Experimentation Ethics Committee (and their care was in accordance with the institutional guidelines). Post-mortem patient tissues were provided by the Neurological Foundation Human Brain Bank with institutional ethics approval #011654 (7 HD patients and 7 unaffected individuals; striatum and cerebellum) directed by Richard L.M. Faull and Maurice A. Curtis. The healthy control iPSCs were derived from human skin biopsy fibroblasts, collected under ethical approval granted by the South Wales Research Ethics Committee (WA/12/0186) in the James Martin Stem Cell Facility, University of Oxford, under standardized protocols.

The HD iPSCs were obtained from NINDS Human Cell and Data Repository. The informed consent was obtained by participants and the human study was conducted in accordance to the criteria set by the Declaration of Helsinki.

## Plasmids

The *pCMV6-PAPD5-myc* (RC229323) plasmid was purchased from Ori-Gene Technologies. The site-directed mutagenesis was performed by GeneWiz (Suzhou, China) to generate the *pCMV6-PAPD5$^{D256A\&D258A}$-myc* plasmid. To generate *pEGFP-STOP-Htt$_{1-550}$CAG91* and *pEGFP-Htt$_{1-550}$Q89$_{(CAA/G)}$* constructs, the *STOP-Htt$_{1-550}$CAG91 and Htt$_{1-550}$Q89$_{(CAA/G)}$* DNA fragments (Supplementary Table 1) were synthesized by GENEWIZ (Suzhou, China) and subcloned into *pEGFP-C1* (Clontech) vector using *Bgl*II and *Kpn*I. To generate *PAPD5$^{+561/+860}$ wild-type-luciferase* and *PAPD5$^{+561/+860}$ YY1 mutant-luciferase* constructs, the *PAPD5$^{+561/+860}$ wild-type* and *PAPD5$^{+561/+860}$ YY1 mutant* DNA sequences were synthesized by GenScript and subcloned into *pGL4.17[luc2/Neo]* firefly luciferase vector (E6721, Promega) using *Nhe*I and *Hin*dIII. To generate *pRI-CMV/GFP-miR-7*, a *miR-7* mimic sequence, 5′-<u>TGGAAGAC TAGTGATTTTGTT</u>-3′, was embedded in an optimized murine *miR-155* scaffold sequence: *miR-7* forward: 5′-TGCTG <u>TGGAAGAC TAGTGATTTTGTT</u> GTTTTGGCCACTGACTGACAACAAAATCTA GTCTTCCA-3′ and *miR-7* reverse: 5′-CCTGTGGAAGACTAGATTTTG TTGTCAGTCAGTGGCCAAAAC<u>AACAAAATCACTAGTCTTCCAC</u>-3′. Synthesized DNA fragments were then subcloned into *pRI-CMV/GFP-miRNA* vector (V6501, Inovogen) at the *Bbs*I restriction site according to the manufacturer's instructions.

## Cell culture and transfection

The human neuroblastoma SK-N-MC cells (HTB-10™, American Type Culture Collection) were cultured in DMEM (11995065, Thermo Fisher Scientific) supplemented with 10% fetal bovine serum (F7524, Sigma-Aldrich) and 1% penicillin–streptomycin (15140122, Thermo Fisher Scientific). The iPSC lines used in this study were described previously[55,64–66]. The demographic information regarding these lines is summarized in Supplementary Table 2. The iPSCs were cultured on Geltrex (A1413302, Thermo Fisher Scientific) using mTeSR1 medium (85850, StemCell Technologies) supplemented with 1% penicillin-streptomycin solution. All cells were maintained in a 37 °C humidified cell culture incubator supplemented with 5% $CO_2$. All cell lines were tested negative for mycoplasma contamination using the MycoAlert™ plus Mycoplasma Detection Kit (LT07-703, Lonza).

## Striatal neuron differentiation

The protocol used to differentiate iPSCs to striatal neurons was described previously[34]. Briefly, the iPSCs were induced in complete N2B27 medium[67] supplemented with 100 nM LDN-193189 (72147, StemCell Technologies), 200 nM dorsomorphin (100-0246, StemCell Technologies) and 10 μM SB431542 (72234, StemCell Technologies). The induction medium was changed every day until day 4. On day 5, the SB431542 was withdrawn, and the cells were maintained in complete N2B27 medium supplemented with LDN-193189 and dorsomorphin until day 9. The cells were then split at 1:2 and maintained in patterning medium (complete N2B27 medium supplemented with 25 ng/mL recombinant human Activin A (338-AC-050/CF, R&D systems) until day 20. On day 21, the neural progenitors were split again and plated at desired densities in culture plates. The medium was changed to complete N2B27 medium supplemented with 10 ng/ml human brain-derived neurotrophic factor recombinant protein (PHC7074, Thermo Fisher Scientific) and 10 ng/ml human glial cell line-derived neurotrophic factor recombinant protein (PHC7045, Thermo Fisher Scientific). The day 35 neurons were used in all experiments. For BCH001 treatment, the BCH001 was added into the medium and was refreshed every other day. A detailed differentiation scheme is presented in Supplementary Fig. 5a. The differentiation for

both healthy control and HD iPSCs was repeated independently for three times.

## Plasmid and siRNA transfection

Plasmid DNA and siRNA transfection were performed using Lipofectamine™ 2000 (11668019, Thermo Fisher Scientific) and Lipofectamine™ RNAiMAX (13778150, Thermo Fisher Scientific), respectively. The transfection was carried out according to the manufacturer's instructions. The ON-TARGETplus SMARTpool siRNAs from Dharmacon (Horizon Discovery Ltd.) were used to knock down the expression of *PAPD5*, *Huntingtin*, and *YY1*. Specifically, 20 pmol of *PAPD5*-siRNA (L-010011-00-0005), 5 pmol of *Huntingtin*-siRNA (L-003737-00-0005), and 5 pmol of *YY1*-siRNA (L-011796-00-0005) were used. Non-targeting siRNA (D-001210-01-50) was used as a control.

## Immunocytochemistry (ICC)

For ICC performed on SK-N-MC cells, cells were fixed with 4% paraformaldehyde for 15 min, followed by permeabilization (0.1% Triton X-100) for another 15 min and blocking (1% BSA) for 1 h at room temperature. Primary antibody incubation was performed in 1% BSA overnight at 4 °C. After washing steps, cells were incubated with secondary antibodies in 1% BSA in dark for 1 h at room temperature. The primary and secondary antibodies used were anti-YY1 (1:200, ab109237, Abcam), anti-polyglutamine (1:200, MAB1574, Merck Millipore), Alexa Fluor 488 Donkey anti-Rabbit IgG (H + L) (1:500, ab150073, Abcam) and Alexa Fluor 594 Donkey anti-Mouse IgG (H + L) (1:500, A-21203, Thermo Fisher Scientific).

For ICC performed on iPSC-derived striatal neurons, neurons were fixed with 4% paraformaldehyde for 15 min, followed by permeabilization (0.2% Triton X-100) for another 15 min and blocking (1% BSA) for 1 h at room temperature. The following steps are the same as described above. The primary antibodies used were anti-YY1 (1:200, ab109237, Abcam), anti-DARPP32 (1:200, ab40801, Abcam), anti-polyglutamine (1:200, MAB1574, Merck Millipore) and anti-MAP2 (1:10,000, ab5392, Abcam). Secondary antibodies used were Alexa Fluor 488 Donkey anti-Rabbit IgG (H + L) (1:500, ab150073, Abcam), Alexa Fluor 594 Donkey anti-Mouse IgG (H + L) (1:500, A-21203, Thermo Fisher Scientific) and Alexa Fluor 647 Goat anti-Chicken IgY (H + L) (1:500, ab150171, Abcam). Samples were counterstained with Hoechst 33342 (1:400, H-1399, Thermo Fisher Scientific) for nuclei visualization. Cell images were acquired on a TCS SP8 high speed imaging system (Leica Microsystems) using a 63x water immersion objective lens. Only representative images are shown.

## Protein sample preparation, western blotting, and antibodies used in this study

Proteins were extracted from SK-N-MC cells or iPSCs-derived striatal neurons using SDS sample buffer. All protein samples were boiled at 99 °C for 10 min, prior to western blotting. Primary antibodies used were anti-PAPD5 (PA5-46747, 1:1000) from Invitrogen; anti-p-TAK1 (4536, 1:1000), anti-t-TAK1 (4505, 1:2000), anti-p-MKK4 (9156, 1:1000), anti-t-MKK4 (9152, 1:2000), anti-p-JNK (9251, 1:2000), anti-t-JNK (9252, 1:2000), anti-TAB2 (3745, 1:2000), anti-cleaved caspase-3 (9664, 1:500) and anti-myc (2276, 1:2000) from Cell Signaling Technology; anti-Huntingtin (MAB5374, 1:1000) from Merck Millipore; anti-YY1 (ab109237, 1:2000) and anti-β-tubulin/dβ-tubulin (ab6046, 1:2000) from Abcam; anti-GFP (632381, 1:4000) from Clontech, and anti-dYY1 (orb806899, 1:1000) from Biorbyt. Secondary antibodies used were goat anti-rabbit (11-035-045, 1:5000) and goat anti-mouse (115-035-062, 1:5000) from Jackson ImmunoResearch. The membrane was washed three times with 1× TBST each for 10 min, followed by chemiluminescent signal detection. The signal was developed using Immobilon Forte Western HRP substrate (WBLUF0100, Merck Millipore), and the images were captured and processed using ChemiDoc™

Touch Imaging System. β-tubulin was used as the loading control. Only representative blots are shown.

## Fluorescence in situ hybridization (FISH) coupled with ICC

Cells were fixed with 4% paraformaldehyde/DEPC-PBS for 15 min and permeabilized with 0.2% Triton X-100/DEPC-PBS for 15 min at room temperature. After washing, cells were incubated in the hybridization buffer (50% formamide, 10% dextran sulfate, 2X saline-sodium citrate (SSC) and 50 mM sodium phosphate buffer) at 65 °C for 20 min. Cells were then hybridized with denatured TYE563-labeled Affinity Plus DNA probe (5′-TYE563-C + TGC + TGC + TGCTG + CTG + CTG + CT-3′; Integrated DNA Technologies) in hybridization buffer for 4 h at 65 °C. The concentrations of the probe used were 40 nM in SK-N-MC cells and 80 nM in iPSC-derived striatal neurons. After hybridization, cells were washed with 2x SSC/0.1% Tween-20 buffer 3 times at 65 °C, followed by further washing with 0.1x SSC buffer for 3 times at 65 °C. Cells were then blocked with 1% BSA in DEPC-PBS for 1 h at room temperature and stained with anti-YY1 (1:200; ab109237, Abcam) overnight at 4 °C. After washing, cells were incubated with Alexa Fluor 488 Donkey anti-Rabbit IgG (H + L) (1:500, ab150073, Abcam) in dark for 1 h at room temperature. Cell nuclei were counterstained with Hoechst 33342. Cell images were acquired on a TCS SP8 high speed imaging system (Leica Microsystems) using a 63x water immersion objective lens. Only representative images are shown.

## Total RNA extraction and quantitative reverse transcription polymerase chain reaction (RT-qPCR)

Total RNA was isolated from SK-N-MC cells and fly heads using RNA extraction kit (9767, Takara Bio Inc.) and TRIzol™ reagent (15596018, Thermo Fisher Scientific), respectively. The reverse transcription (RT) was performed using the ImProm-II™ Reverse Transcription System (A3803, Promega) according to the manufacturer's instructions. The TB Green master mix (RR420A, Takara Bio Inc.) was used for real-time PCR (qPCR). Thermal cycling was performed on Bio-Rad CFX96 Real-time PCR detection system (Bio-Rad Laboratories) under the following conditions: 10 min at 95 °C, and 45 cycles of 15 s at 95 °C and 1 min at 60 °C. The $\beta$-actin was used as the internal control ($Ct_{\beta\text{-actin}}$), and relative gene expression was quantified using the $2^{-\Delta\Delta CT}$ method. The $\Delta Ct = Ct_{\text{target gene}} - Ct_{\beta\text{-actin}}$. The $\Delta Ct$ values from three sets of control samples were averaged to obtain the $\Delta Ct_{\text{control-averaged}}$, and Ct values from each set of control and experimental group samples were further normalized to $\Delta Ct_{\text{control-averaged}}$ to obtain the $\Delta\Delta Ct$ values ($\Delta\Delta Ct = Ct - \Delta Ct_{\text{control-averaged}}$). Primers used in this study were listed in Supplementary Table 4.

## Small RNA extraction and qPCR screening

Small RNA species from SK-N-MC cells were purified using RNeasy Mini (74104, Qiagen) and RNeasy MinElute Cleanup (74204, Qiagen) kits, followed by reverse transcription using miRCURY® LNA® RT system (339340, Qiagen) according to the manufacturer's instructions. Samples were then subjected to miRCURY LNA miRNA miRNome PCR Panels (YAHS-312YE-8; Qiagen) for human miRNAs screening under miRCURY® LNA® miRNA real-time PCR (Qiagen) system, following the manufacturer's instructions. The *U6 snRNA* was used as the internal control ($Ct_{\text{U6 snRNA}}$), and relative miRNA expression was analyzed using the $2^{-\Delta\Delta CT}$ method. The $\Delta Ct = Ct_{\text{target miRNA}} - Ct_{\text{U6 snRNA}}$. The $\Delta Ct$ values of *miR-7-5p* miRNA from three sets of "Untransfected + control-siRNA" samples were averaged to obtain the $\Delta Ct_{\text{averaged}}$, and Ct values from each set of control and experimental group samples were further normalized to $\Delta Ct_{\text{averaged}}$ to obtain the $\Delta\Delta Ct$ values ($\Delta\Delta Ct = Ct - \Delta Ct_{\text{averaged}}$). The 186 miRNAs manipulated by PAPD5 in *EGFP$_{CAG78}$*-expressing cells are listed in Supplementary Data 1.

## Functional analysis of miRNAs target genes

Target prediction of miRNAs was performed using miRDB (5.0)[68] and TargetScan (7.2)[69]. A total of 7158 target genes were identified by both miRDB and TargetScan. The intersection of target genes predicted by both databases were used for functional analysis using KOBAS[70]. We focused on the top ten enriched KEGG pathways, which are mainly related to cell survival.

## Small RNA-seq and analysis

The small RNA species were isolated from SK-N-MC cells using the *mir*Vana™ miRNA Isolation Kit (AM1560, Thermo Fisher Scientific) following the manufacturer's instructions. Both library preparation and SE50 sequencing on Illumina platform were performed by Novogene (Beijing, China). Raw sequencing files were checked using FastQC v0.11.8[71]. Adapter and quality trimming were performed using Cutadapt v3.7[72]. FastQC was performed on the trimmed data again to confirm the removal of adapter sequence. The reads (canonical counts) were aligned to mature human miRNAs retrieved from miRbase release 22.1[73] using Bowtie v1.3.1[74]. The reads were also mapped to a custom reference with an "A" added to each miRNA using Bowtie (adenylated counts). SAMtools v1.13[75] was used to retrieve the canonical and adenylated counts. miRNAs with adenylated counts >100 across samples were further processed to calculate their adenylation levels using the following equation:

$$\text{Adenylation level (\%)} = \left( \frac{\text{adenylated counts}}{\text{adenylated counts} + \text{canonical counts}} \right) \times 100\% \quad (1)$$

The statistical analysis was performed using one-way ANOVA followed by *post hoc* Fisher's LSD test. The adenylation level of processed miRNAs is listed in Supplementary Data 2.

## Detection of adenylated *miR-7-5p* level using RT-qPCR

The detection of adenylated *miR-7-5p* level was carried out as described previously[14,76]. Briefly, small RNAs from SK-N-MC cells following the treatment of control-siRNA or *PAPD5*-siRNA were isolated using the *mir*Vana™ miRNA Isolation Kit. The circularization of miRNAs was performed using the T4 RNA ligase (M0204S, New England Biolabs) and 250 ng small RNAs as input. Two microliters of each circularization reaction were used for the RT. The RT primer used was 5′-CTA GTCTTCCATA-3′. The qPCR primers were: 5′-GGAAGACTAG TGATTTTGTTGTTAT-3′ and 5′-CAACAAAATCACTAGTCTTCCATAA-3′. The relative level of adenylated *miR-7-5p* was analyzed using the $2^{-\Delta Ct}$ method. The Ct values of adenylated *miR-7-5p* from three sets of "control-siRNA" samples were averaged to obtain the $Ct_{\text{averaged}}$, and Ct values from each set of control and experimental group samples were further normalized to $Ct_{\text{averaged}}$ to obtain the $\Delta Ct$ values ($\Delta Ct = Ct - Ct_{\text{averaged}}$).

## Collection and preparation of human brain tissue samples

The brain tissue (striatum and cerebellum) from the unaffected individuals and HD patients were collected and processed as described previously[35]. The detailed clinical information is summarized in Supplementary Table 3. Total RNA was isolated from brain tissues using RNeasy Mini Kit. The RT was performed using the ImProm-II™ Reverse Transcription System according to the manufacturer's instructions. Small RNA species were isolated using RNeasy Mini and RNeasy MinElute Cleanup Kits following the manufacturer's instructions. Enriched small RNAs were polyadenylated using poly(A) polymerase (AM1350, Thermo Fisher Scientific) in the presence of $MnCl_2$ (2.5 mM) and ATP (1 mM) at 37 °C for 1 h. PolyA-tailed small RNAs were then annealed with a polyT-enriched RT adapter (5′- CGAATTCTAG AGCTCGAGGCAGGCGACATGGCTGGCTAGTTAAGCTTGGTACCGAGC TCGGATCCACTAGTCCTTTTTTTTTTTTTTTTTTTTTTTTTTAC-3′), followed by the RT reaction. The target genes and miRNAs were amplified using the primers listed in Supplementary Table 4. The *β-actin* and *RNU66* were used as the internal controls, and relative gene expression was quantified using the $2^{-\Delta\Delta CT}$ method.

The protein samples for immunoblotting were prepared as described previously[35]. The primary antibodies used were anti-PAPD5 (1:1000, 55197-1-AP, Proteintech), anti-TAB2 (1:1000, 3745, Cell Signaling Technology) and anti-Actin (1:30,000, 612657, BD Transduction Laboratory). Secondary antibodies used were goat anti-rabbit (G-21234, 1:2500) and goat anti-mouse (G-21040, 1:2500) from Thermo Fisher Scientific.

## ChIP-seq

The genomic DNA was isolated from cells using the Pierce™ Magnetic ChIP Kit (26157, Thermo Fisher Scientific). The experimental procedures were carried out following the manufacturer's instructions. Four micrograms of anti-YY1 (22156-1-AP, Proteintech) antibody were used for the IP. Both library preparation and PE150 sequencing on Illumina platform were performed by Novogene (Beijing, China). Raw sequencing files were checked using FastQC v0.11.8[71]. Adapter and quality trimming were performed using Cutadapt v3.7[72]. FastQC was performed on the trimmed data again to confirm the removal of adapter sequence. The reads were then aligned to human genome GRCh38.p14 using Bowtie2 v2.4.1[77]. Uniquely aligned reads were retrieved using Sambamba v1.0.1[78], and those overlapping with regions in the Encyclopedia of DNA Elements (ENCODE) blacklist[79] were removed using BEDTools v2.29.2[80]. The YY1-binding DNA sites (peaks) were identified using MACS3 v3.0.1[81] with input control as background, annotated using ChIPseeker v1.38.0[82], and visualized using Trackplot v1.5.10[83]. Read counts were obtained following the normalization to sequencing depth using DiffBind v3.12.0[84]. The peaks with read counts more than zero in each of the samples were further processed. Promoter was defined as ± 2 kilobases from a gene's transcription start site (TSS)[85]. The YY1-targeted genes were selected according to the ENCODE project[51,86]. The YY1-targeted genes that show reduced binding to YY1 in *EGFP-Htt$_{1-550}$Q89*-expressing cells are listed in Supplementary Data 3. Statistical analysis was performed using one-tailed unpaired Student's *t*-test.

## Dual-luciferase reporter assay

The dual-luciferase reporter assay (E1910, Promega) was carried out as described previously[31]. The *pTK-RL Renilla* luciferase vector (E2241, Promega) was used as an internal control reporter construct for the normalization of transfection efficiency. Both firefly and *Renilla* luminescence were recorded on a Spark® multimode microplate reader (Tecan). The relative luciferase activity was calculated by dividing the firefly luminescence reading by the *Renilla* luminescence reading.

## Lactate dehydrogenase (LDH) assay

The LDH assay on SK-N-MC cells were performed according to manufacturer's instructions using the CytoTox 96 Non-Radioactive Cytotoxicity Assay Kit (G1780, Promega). The absorbance of LDH enzyme released into the culture medium (Absorbance$_{supernatant}$) and from attached cells (Absorbance$_{attached}$) were measured on a Spark® multimode microplate reader (Tecan, Switzerland). The relative cell death was calculated using the following equation.

$$\text{Relative cell death} = \frac{\text{Absorbance}_{supernatant}}{\text{Absorbance}_{supernatant} + \text{Absorbance}_{attached}} \quad (2)$$

## Drosophila stocks

Fly lines *GMR-GAL4*[87], *UAS-DsRed$_{CAG0/100}$*[3], *UAS-Htt$_{exon1}$Q93*[28], *UAS-CTG$_{60/480}$*[88] and *UAS-CGG$_{90}$-EGFP*[89] were described previously. The *UAS-PAPD5 dsRNA* line (GD19799) and *UAS-dYY1 dsRNA* line (GD39529) were obtained from Vienna *Drosophila* RNAi Center. The *UAS-dYY1* line (F000151) and *Tub56D$^{Gene-Switch}$-GAL4* line (40333) were obtained from FlyORF and Bloomington *Drosophila* Stock Center, respectively. All flies were maintained in cornmeal culture medium and genetic crosses were set up in a 21.5 °C incubator (LMS). Both male and female flies were used in all experiments.

## External eye examination

Adult flies were anesthetized with $CO_2$ and examined using an Olympus SZX12 stereomicroscope. External eye images were captured by a SPOT Insight CCD camera and all the images were processed using the SPOT Advanced software and Adobe Photoshop 2020 software. A total of 10 fly eyes from 10 adult flies were examined under the microscope for each condition.

## Pseudopupil assay

Adult flies were used for the pseudopupil assay as described previously[90]. Each fly eye was examined using an Olympus CX31 stereomicroscope. The images of ommatidia were captured using a SPOT Insight CCD camera (Diagnostic instruments Inc.), and all images were processed using the SPOT Advanced software (Version 5.2; Diagnostic instruments Inc.) and Photoshop 2020 (Adobe Systems). The average number of rhabdomeres per ommatidia was presented as the indication of ommatidial integrity. A total number of 600 rhabdomeres from 30 flies were examined under each control or experimental condition over three independent experiments. The age of flies examined were 18 days post eclosion (dpe) for *DsRed$_{CAG0}$* and *DsRed$_{CAG100}$* flies, and 3 dpe for *Htt$_{exon1}$Q93* flies.

## BCH001 treatment

Fly third instar larvae were fed with 10 μM of BCH001 (dissolved in 2% sucrose solution) for 2 h at room temperature. After feeding, larvae were cultured in standard fly food until being used for pseudopupil assay.

## Climbing assay

The 0-3 dpe flies were transferred to fly food containing 200 μM mifepristone (RU486, ab120356, Abcam). The RU486 was used to induce transgene expression. The climbing ability of flies was measured by negative geotaxis assay. Flies from each experimental group were anesthetized and placed in a vertical plastic column. After an hour of recovery in 25 °C incubator, flies were banged to the bottom, and the time they used to climb across 6 cm in 30 s was recorded. For those flies failed to climb across 6 cm in 30 s, the climbing time was marked as 30 s. The ages of flies used are 22–25 dpe.

## Software

The intensity of the protein bands were quantified using the ImageJ software 1.52k (https://imagej.nih.gov/ij/)[91]. To perform the quantification analyses, the band intensities from control samples were averaged to obtain the averaged value, all values from control and experimental groups were further expressed relative to that average. Confocal images were analyzed using the Leica Application Suite X software. All absorbance and luminescence readings were determined by the SparkControl software version 2.1. The bar charts were plotted using GraphPad Prism version 10.0.3. The illustrations were created using Adobe Illustrator 27.0.1 (Figs. 1m, p, 3c, k, 6a) and BioRender.com (Figs. 8b, 10 and Supplementary Fig. 5a). Assembly of panels in finalized figures was done using Adobe Illustrator 27.0.1.

## Statistical analysis

The unpaired Student's *t* test (one-tailed or two-tailed) was used for the comparison between two experimental groups. One-way ANOVA followed by *post hoc* Tukey's test or one-way ANOVA followed by *post hoc* Fisher's LSD test was applied when comparing three or more groups. *, **, *** and **** represent $P < 0.05$, $P < 0.01$, $P < 0.001$ and $P < 0.0001$, respectively, which are considered statistically significant. ns indicates no significant difference. The violin plots show the median (dashed lines), and 25th and 75th quartiles (dotted lines).

GraphPad Prism version 10.0.3 was used to plot data and perform statistical analysis.

## Reporting summary

Further information on research design is available in the Nature Portfolio Reporting Summary linked to this article.

## Data availability

The data supporting the findings of this study are available from the corresponding authors upon request. Transcription factor binding sites were predicted using Transcription factor affinity prediction (http://trap.molgen.mpg.de/cgi-bin/trap_form.cgi)[92], PROMO (http://alggen.lsi.upc.es/cgi-bin/promo_v3/promo/promoinit.cgi?dirDB=TF_8.3.)[93] and JASPAR[94] databases. The human, rat, mouse, and fly *PAPD5* promoter sequences were withdrawn from GenBank under accession codes NC_000016.10 [https://www.ncbi.nlm.nih.gov/nuccore/NC_000016.10?from=50152911&to=50235310&report=genbank], NC_086037.1 [https://www.ncbi.nlm.nih.gov/nuccore/NC_086037.1?from=34984244&to=35042423&report=genbank&strand=true], NC_000074.7 [https://www.ncbi.nlm.nih.gov/nuccore/NC_000074.7?from=88925229&to=88989942&report=genbank], and NT_033777.3 [https://www.ncbi.nlm.nih.gov/nuccore/NT_033777.3?from=24925216&to=24927085&report=genbank], respectively. The raw cycle threshold values for miRNA array have been deposited in GEO under accession code GSE271666. The raw sequencing files for small RNA-seq and ChIP-seq have been deposited in GEO under accession codes GSE272903 and GSE273082, respectively. Source data for the figures and Supplementary Figs. are provided as a Source Data file. Source data are provided with this paper.

## Code availability

Any codes used for the small RNA-seq and ChIP-seq data analyses are available upon reasonable request to the corresponding author (hyechan@cuhk.edu.hk).

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

## Acknowledgements

We acknowledge the Neurological Foundation Human Brain Bank for brain tissues from HD patient and unaffected individuals. This work was supported by Research Grants Council General Research Fund (Project Number 14107118); CUHK Vice-Chancellor's One-Off Discretionary Fund (Project Numbers 4930713 and VCF2014011) and Gerald Choa Neuroscience Centre (Project Number 7105306), Faculty of Medicine, The Chinese University of Hong Kong. K.M.K. and T.-F.C. were supported by the Innovation and Technology Commission, Hong Kong Special Administrative Region Government to the State Key Laboratory of Agrobiotechnology (CUHK). Any opinions, findings, conclusions, or recommendations expressed in this publication do not reflect the views of the Government of the Hong Kong Special Administrative Region or the Innovation and Technology Commission.

## Author contributions

Z.S.C., S.I.P., L.I.L., T.G.-D., N.S.J.W., T.H.L., X.L., Y.W., A.C.K., J.H. and J.K.-L.S. conducted the experiments and performed the data analyses. C.T., L.T., M.A.C., and R.L.M.F. provided the patient brain samples. K.T. provided the healthy control human iPSC lines. K.M.K., H.-M.C., H.K., T.-F.C., C.E.P., and H.Y.E.C. supervised the research. The manuscript was written by Z.S.C., S.I.P. and H.Y.E.C. and commented on by all authors.

## Competing interests

The authors declare no competing interests.
