## [Transparent Peer Review file · Nature Communications]

Mutant huntingtin induces neuronal apoptosis via derepressing the non-canonical poly(A) polymerase PAPD5

Corresponding Author: Professor Ho Yin Edwin Chan

Version 0:

Reviewer comments:

Reviewer #1

(Remarks to the Author)

In this manuscript, mostly based on ectopic overexpression models of CAG repeats, the authors identified a signaling cascade involving YY1, PAPD5, two microRNAs, TAB1/2, TAK1, MKK4, and JNK as a potential mediator of both polyQ protein and RNA toxicities in Huntington's disease (HD). They argued that YY1 was recruited to RNA foci containing CAG repeats and polyQ-containing protein aggregates, leading to loss of YY1 function due to sequestration. This loss of functional YY1 resulted in the up-regulation of PAPD5 and a decrease in its miRNA targets, subsequently activating the TAB1/2-mediated TAK1–MKK4–JNK pro-apoptotic pathway. The authors also demonstrated that inhibitors targeting PAPD5, TAK1, or JNK could alleviate cell death in both cell and fly models of CAG repeats. While some aspects of the study seem to be convincing, critical gaps in this proposed pathway remain. Furthermore, the relevance of these findings to HD is far from established, which severely limit the significance of these results.

Major concerns:

1. Disease relevance. Nearly all experiments were performed in human cancer cell line and drosophila ectopic expression models. The one experiment they performed in iPSC-derived cortical neurons (Fig.3D) was woefully underpowered (N=1). Critically missing from this study is the analysis of HD patient samples. Do the expression levels of PAPD5 and its target miRNAs consistently change in HD patients compared to healthy individuals? Do these changes occur specifically in striatum, which is mostly affected in HD?
2. Role of PAPD5 in miRNA degradation. The extent to which PAPD5 KD affected the abundance of nearly all miRNAs (Fig.2A) is surprising. Even if PAPD5 is essential for TDMD, only a small subset of miRNAs are subject to TDMD (e.g., PMID: 36150386). To establish the role of PAPD5 in the degradation of miR-504 and miR-7, the authors need to show that their adenylation requires PAPD5.
3. Sequestration of YY1. In overexpression models, while YY1 colocalized with polyQ aggregates and CAG RNA foci (Fig. 6), the soluble and diffuse YY1 level was not reduced compared to neighboring cells lacking RNA foci or polyQ aggregates, which simply rejects the hypothesis of any sequestration. To confirm that YY1 is indeed sequestered, the authors should demonstrate a decrease in the binding of YY1 to the promoter regions of PAPD5 and other YY1 target genes using CHIP-seq. More importantly, can this “sequestration” be observed in patient cells?
4. The authors observed increased PAPD5 expression and activation of the TAK1-MKK4-JNK signaling pathway in HD patient iPSC-derived neurons. The authors should investigate whether perturbing the proposed pathway can suppress cell death in iPSC-derived neurons. Additionally, it is essential to test whether YY1 is recruited to endogenous RNA foci or polyQ aggregates in these cells.

Reviewer #2

(Remarks to the Author)

In this manuscript, Peng et al. identified PAPD5 (TENT4B) as an essential factor in cell toxicity observed in Huntington disease (HD). Through a set of studies, they established a model in which expression of pathogenic CAG-repeat RNAs induces PDPD5 upregulation which in turn triggers pro-apoptotic signaling pathway via downregulation of miRNAs.

The conclusion that PAPP5 plays a critical role in CAG-RNA-mediated cell toxicity is convincing. Not only the authors showed that inhibiting PAPP5 function by siRNA knockdown or by a small molecular drug rescues cell death in both cell-based and drosophila HD models, but they also revealed sequestering YY1, a negative regulator of PAPP5 transcription, as the mechanism of PAPP5 upregulation in HD cells. Interestingly, knocking down PAPP5 in WT cells apparently does not affect cell growth, making PAPP5 a potential therapeutic target for HD.

On the contrary, authors' claim that PAPP5 mediates cell-death via miRNAs is improbable and not supported by the data. Authors showed that manipulating certain miRNAs induces or alleviate the observed phenotype. However, this only implicates miRNAs in apoptosis pathway, which is somewhat expected. A direct link between PAPP5 and miRNA downregulation is required but nonetheless missing in this study. Previous reports only established PAPP5 in degradation of a specific isoform of miR-21. A global downregulation of miRNAs was observed in HD cells, indicating that it is unlikely to be mediated by PAPP5 directly but rather an indirect result of cell toxicity. Given that PAPP5 promotes degradation of telomerase RNA, it is possible that PAPP5 promotes cell apoptosis via dysregulation of telomerase activity.

In addition, this study will benefit from including additional controls such as catalytic-inactive PAPP5 mutant. I cannot recommend publishing this manuscript in its current form.

Major concerns:

1. Line 67-69. Authors state that PAPP5 targets miRNAs for degradation via 3' adenylation. This is not the consensus of the field. It is misleading to claim that PAPP5 is known to be involved in miRNA degradation in general. Ref 11 reports a specific case while ref12 does not show any evidence of PAPP5-mediated miRNA degradation.
2. Fig 1B. It is interesting that PAPP5 KD in non-induced cells showed no impact on cell death. What are the levels of PAPP5 after siRNA KD? A western blot should be provided here.
3. Fig 1F/G. To rule out the possibility that induced apoptosis is non-specific (caused by overexpression of non-specific proteins), authors should include catalytic-inactive PAPP5 mutant as a control. Authors are also encouraged to use this mutant in following experiments whenever possible to demonstrate whether observed effect is dependent on the adenylation activity.
4. Fig 2A. How were miRNA levels measured and how is this list of 184 miRNAs determined? Are these highly expressed miRNAs in this cell type? In addition to heatmap showing fold-changes, miRNA expression levels should be provided.
5. To examine whether PAPP5 directly targets these downregulated miRNAs, authors should perform NGS and measure the changes on 3' adenylation of these miRNAs.
6. Fig 5E. Authors should include the mutant promoter luc-reporter here.
7. Line338. I am not aware that PAPP5 is implicated in TDMD. Please provide refs.

Minor points:

1. According to HUGO-approved nomenclature, PAPP5 should be referenced as TENT4B. This should be at least mentioned in introduction.
2. Figure and figure legends should provide essential information on how to interpret the results shown, rather than merely repeat the conclusions which are already stated in the main text. Using Fig1A as an example, it should be clearly indicated that the upper panel is a gel of RT-PCR and the bottom gel is a Western blot either in the figure itself or in the legend.

Reviewer #3

(Remarks to the Author)

This study describes a comprehensive analysis of the regulatory network leading to the mis-regulation of miRNAs in Huntington disease's (HD) fly and cell models and how this leads to cell death. The authors focus on 2 miRNAs, miR-504 and miR-7, showing how these are downregulated in an RNA-only model of HD and how this downregulation leads to the activation of a TAK1-MKK4-JNK pro-apoptotic signalling cascade. The authors go on to show that the downregulation is modulated by the increase in PAPP5, and go on to show how the transcription factor YY1 suppresses PAPP5 expression and how in protein models of HD there is an upregulation of PAPP5. They propose a model where YY1 protein is recruited to CAG RNA foci and polyQ protein aggregates, leading to the de-repression of PAPP5 transcription. As a consequence, PAPP5 expression increases, causing the degradation of miR-504 and miR-7. This, in turn, initiates the TAK1-MKK4-JNK-caspase cascade, eventually triggering caspase 3 activation and subsequent neuronal cell death.

This study is well written, presents a wealth of good quality data. However there are some concerns that need to be addressed:

- 1-None of the western blots (apart from Fig 6) are quantified. The standard for the field is to quantify replicates, display error bars, and describe differences with appropriate statistical analysis. This should be done for all western blots displayed.
- 2-Similarly for mRNA analysis throughout, this should be done by qPCR, not by showing a single, unquantified, sample
- 3-In all graphs displaying relative cell death the control samples can't be set to 1 as this eliminated any error bars from the control sample, thus prejudicing the analysis (an ANOVA can't be use if the samples have no error). It is fine to calculate the mean of the control sample, set that value to 1 but then all values need to be expressed relative to that average (including

the control samples), so as to retain error bars for all samples.

4-Similarly for the quantification for YY1 in Fig 6, it is good that the western blots are quantified, but amounts either need to be expressed as quantified values, or normalised to the average of the control sample, as described above, but error bars have to be present for the control sample

5- Can the number of rhabdomeres/ flies quantified please be annotated in each figure.

6-Could the authors please comment as to why in Fig 5F and 5H, in the control situations where YY1 is down-regulated, there is no increase in PAPD5?

7-Can the authors please comment as to why in Fig 5R in the UAS-Httexon1 Q93 over-expressing flies, there is no decrease in YY1 transcript but in Fig 6A the same flies show a reduced YY1 expression. Does that mean YY1 is regulated at the level of translation?

8-In Fig 5G and 5I it would be good to see a western blot with a YY1 antibody (rather than myc), to show the effect of the mutations on the endogenous YY1 and how much is YY1 increased with the over-expression of YY1-myc.

9-In Fig 5H there should a drop of YY1 in the EGFP-Htt1-550 Q89(CAG) condition, why is it not? The authors should comment

10- The co-localisation in Fig 6E and 6G should be analysed for statistical significance, for example with a Pearson correlation coefficient

11-To formally prove that PAPD5 and YY1 are in a linear pathway and not contributing to the disease formation separately from each other, it would good to show that RNAi of PAPD5, together with over-expression of YY1 does not lead to additive effects, the two interventions should be epistatic to each other. This experiment could easily be done in the fly eye model

12-The authors carry out a comprehensive analysis using a fly eye as disease model, however this is very much a developmental phenotype, whereas HD is an adult-onset disease. The disease relevance of the paper would be much greater if the authors showed the effects of PAPD5 down-regulation on an adult phenotype for the fly HD model (lifespan/climbing/activity). This is not essential.

Version 1:

Reviewer comments:

Reviewer #1

(Remarks to the Author)

Most of the issues in the previous manuscript has been addressed by the additional data and analyses. There are two concerns regarding the newly added data that should be addressed before publication:

1. To show that PAPD5 is required for the miR-7 adenylation, they need to show PAPD5 loss-of-function effects on miRNA adenylation and abundance, rather than overexpression. The author should also show the global analysis of their miRNA array and sequencing data (e.g., volcano plot) and point out the miRNAs of interest.

2. The ChIP-seq quantification says "read counts". These read counts should be normalized to sequencing depth. In addition, they should show some form of global analysis and point out their genes of interest.

Reviewer #2

(Remarks to the Author)

I appreciate the extensive efforts made by the authors to address my initial concerns. The addition of the PAPD5 catalytic mutant experiments has significantly strengthened the manuscript, providing clearer insights into the role of PAPD5's adenylation activity in the observed cellular effects.

However, my primary concern regarding the direct involvement of PAPD5 in the downregulation of miRNAs in Huntington's Disease remains unresolved. The authors have indeed shown that only a subset of the downregulated miRNAs (if not only miR-7) is directly adenylated by PAPD5, which aligns with my original assertion that the widespread miRNA downregulation is more likely an indirect consequence of cellular toxicity rather than a direct effect of PAPD5 upregulation. Hence, the manuscript still lacks clarity on how PAPD5 contributes to the pathology of HD, a key point of this study. Therefore, I am unable to recommend this manuscript for publication in its current form.

Reviewer #3

(Remarks to the Author)

The authors have addressed all my comments. This manuscript is now ready for publication.

Version 2:

Reviewer comments:

Reviewer #1

(Remarks to the Author)

The authors made an attempt to address my specific concerns. However, neither unbiased high-throughput measurements strongly supported their initial findings. The effect of PAPD5 KD on miR-7 abundance is so small (Fig.3g) that it could not possibly have any impact on target repression. A similar issue applies to their genome-wide YY1 ChIP-seq analysis, in which YY1 binding to PAPD5 promoter did not change substantially. This is not a request for additional analyses- the amount of analysis already makes the paper nearly impossible to read- but rather, I suggest the authors think carefully about their model and remove some of the components that their data cannot fully justify.

POINT-BY-POINT RESPONSE TO REVIEWERS' COMMENTS

Reviewer #1 (Remarks to the Authors):

Major concerns:

1. Disease relevance. Nearly all experiments were performed in human cancer cell line and drosophila ectopic expression models. The one experiment they performed in iPSC-derived cortical neurons (Fig.3D) was woefully underpowered (N=1). Critically missing from this study is the analysis of HD patient samples. Do the expression levels of PAPD5 and its target miRNAs consistently change in HD patients compared to healthy individuals? Do these changes occur specifically in striatum, which is mostly affected in HD?

Response: We greatly appreciate the reviewer for pointing out these issues. We have obtained iPSC lines from three healthy individuals (N = 3) and three HD patients (N = 3). All six iPSCs were differentiated into striatal neurons and such differentiation was performed independently for three times (Supplementary Fig. 4a–c). We found the PAPD5 protein level was increased in HD striatal neurons when compared to the controls (Fig. 9c, 9d).

We have analyzed human striatal samples from seven HD patients and seven age-matched unaffected controls (Supplementary Table 3; Gall-Duncan *et al.*, Cell, **186**(22): 4898-4919). When compared with the controls, the PAPD5 protein level was significantly increased (~4.5 fold) in HD striatal tissues (Fig. 9h, 9i). Additionally, the expression of PAPD5-modulated miRNA (*miR-7-5p*) was significantly reduced (Fig. 9j), and its target (TAB2 protein) was significantly upregulated in patients' striata (Fig. 9k, 9l). Our results demonstrate a disease relevance of PAPD5 dysregulation in HD.

We also investigated cerebellar PAPD5 dysregulation in HD (Supplementary Table 3). We detected a milder upregulation (~2-fold) of PAPD5 protein in HD cerebella when compared with the unaffected controls (Fig. 9n, 9o). Consistent with a milder increase of PAPD5 in the cerebellum, we did not observe any alteration of *miR-7-5p* expression and TAB2 level in the cerebellar tissues (Fig. 9p–r). In summary, PAPD5 level is increased moderately in the striatum and mildly in the cerebellum in HD patients, with downstream miRNA dysregulation only being detected in the striatum. Our findings are in line with the fact that striatum is mostly affected, while the cerebellum is minimally affected in HD (Waldvogel *et al.*, Curr Top Behav Neurosci, **22**: 33-80).

2. Role of PAPD5 in miRNA degradation. The extent to which PAPD5 KD affected the abundance of nearly all miRNAs (Fig.2A) is surprising. Even if PAPD5 is essential for TDMD, only a small subset of miRNAs are subject to TDMD (e.g., PMID: 36150386). To establish the role of PAPD5 in the degradation of miR-504 and miR-7, the authors need to show that their adenylation requires PAPD5.

Response: We are grateful to the reviewer for his/her insightful comments. We have performed a miRNA qPCR array (miRCURY LNA miRNA miRNome PCR Panels; YAHS-312YE-8; Qiagen) to assess the expression levels of a total number of 751 human miRNAs (Raw data has been deposited to GEO under the accession number GSE271666). Out of the 751 human miRNAs examined, 186 miRNAs showed downregulation of expression in our mutant CAG RNA-expressing cell model, and their levels could all be restored upon *PAPD5* knockdown (Supplementary Data 1). We also performed small RNA-seq to measure the adenylation level of these miRNAs (Raw data has been deposited to GEO under the accession number GSE272903). Among the 186 miRNAs examined, the adenylation of 11 miRNAs, including *miR-7-5p*, was mediated through PAPD5 (Fig. 3e and Supplementary Fig. 5).

We further showed the adenylation level of *miR-7-5p* was increased in cells overexpressing wild-type PAPD5 protein, but such effect was not observed in cells overexpressed with the PAPD5^{D256A & D258A} catalytic-inactive mutant (Fig. 3e). This indicates PAPD5's catalytic activity is essential for the adenylation of *miR-7-5p*. Regarding *miR-504-5p*, its adenylation level was increased in PAPD5-overexpressing cells. However, the difference did not reach statistical significance level ($P = 0.1449$; Response Fig. 1). Taken together, our new data show that PAPD5 modulates the adenylation of a subset of miRNAs.

Response Fig. 1 The *miR-504-5p* shows an increasing trend of adenylation in wild-type PAPD5-expressing cells when compared to the untransfected cells. Statistical analysis was performed using one-way ANOVA followed by *post hoc* Fisher's LSD test.

3. Sequestration of YY1. In overexpression models, while YY1 colocalized with polyQ aggregates and CAG RNA foci (Fig. 6), the soluble and diffuse YY1 level was not reduced compared to neighboring cells lacking RNA foci or polyQ aggregates, which simply rejects the hypothesis of any sequestration. To confirm that YY1 is indeed sequestered, the authors should demonstrate a decrease in the binding of YY1 to the promoter regions of PAPD5 and other YY1 target genes using ChIP-seq. More importantly, can this “sequestration” be observed in patient cells?

Response: We thank the reviewer for raising his/her concerns. We have performed ChIP-seq to determine the binding of YY1 to the *PAPD5* gene promoter and other YY1 target genes (Raw data has been deposited to GEO under the accession number GSE273082). We found that, when compared to the unexpanded HttQ23-expressing cells, the binding of YY1 to *PAPD5* promoter was significantly reduced in the expanded HttQ89-expressing cells (Fig. 8h, 8i). Similar reduction was also detected in additional YY1 target genes (Supplementary Data 3). The ChIP-seq data support our hypothesis that there is a decrease in the binding of YY1 to the promoter regions of *PAPD5* and other YY1 targets. Furthermore, we show that endogenous YY1 protein co-localized with the CAG RNA foci and polyQ protein aggregates in HD iPSCs-derived striatal neurons (Fig. 9a, 9b). The PAPD5 protein level was increased in these neurons (Fig. 9c, 9d).

4. The authors observed increased PAPD5 expression and activation of the TAK1-MKK4-JNK signaling pathway in HD patient iPSC-derived neurons. The authors should investigate whether perturbing the proposed pathway can suppress cell death in iPSC-derived neurons. Additionally, it is essential to test whether YY1 is recruited to endogenous RNA foci or polyQ aggregates in these cells.

Response: We appreciate the reviewer’s insightful comments. We have obtained iPSC lines derived from three healthy individuals and three HD patients (Supplementary Table 2), and differentiated them into striatal neurons (Supplementary Fig. 4a–c). We observed that endogenous PAPD5 protein level was significantly increased in disease neurons (Fig. 9c, 9d). We also found that the PAPD5-mediated apoptotic pathway was activated in HD iPSCs-derived striatal neurons (Fig. 9e, 9f). We next treated these iPSCs-derived striatal neurons with BCH001, a PAPD5 enzymatic inhibitor (Nagpal *et al.*, *Cell Stem Cell*, 2020, **26**(6): 896–909), to investigate whether perturbing the function of PAPD5 can suppress the activation of this apoptotic pathway in HD. Our results show that the levels of hyperphosphorylated MKK4 and JNK, as well as caspase-3 cleavage were all reduced in BCH001-treated HD striatal neurons (Fig. 9e, 9f). These findings highlight the perturbation of the YY1-PAPD5 regulatory axis and the activation of PAPD5-mediated apoptotic pathway in HD.

We have also performed co-localization experiments and demonstrated that endogenous YY1 protein co-localized with both the endogenous RNA foci (Fig. 9a) and Htt-polyQ protein aggregates (Fig. 9b) in HD striatal neurons.

Reviewer #2 (Remarks to the Authors):

Major concerns:

1. Line 67-69. Authors state that PAPD5 targets miRNAs for degradation via 3' adenylation. This is not the consensus of the field. It is misleading to claim that PAPD5 is known to be involved in miRNA degradation in general. Ref 11 reports a specific case while ref12 does not show any evidence of PAPD5-mediated miRNA degradation.

Response: We apologize for the misleading information and thank the reviewer for his/her insightful comments. The description in the "Introduction" section has been revised and references have been updated (Lines 71 to 75).

2. Fig 1B. It is interesting that PAPD5 KD in non-induced cells showed no impact on cell death. What are the levels of PAPD5 after siRNA KD? A western blot should be provided here.

Response: We thank the reviewer for pointing this out. We have included western blot data in Fig. 1d. Our quantification analysis shows that the PAPD5 protein level was reduced by ~2.5-fold in *PAPD5*-siRNA-treated cells (Fig. 1e).

3. Fig 1F/G. To rule out the possibility that induced apoptosis is non-specific (caused by overexpression of non-specific proteins), authors should include catalytic-inactive PAPD5 mutant as a control. Authors are also encouraged to use this mutant in following experiments whenever possible to demonstrate whether observed effect is dependent on the adenylation activity.

Response: We are grateful to the reviewer for the insightful comments. We included the PAPD5^{D256A & D258A} catalytic-inactive mutant as a control in our study. When overexpressed to a similar protein level (Fig. 1j), we found that only the wild-type PAPD5 would trigger caspase-3 cleavage (Fig. 1j, 1k) and cell death (Fig. 1l). We also performed small RNA-seq on wild-type and mutant PAPD5-expressing cells to further investigate the role of the catalytic function of PAPD5 in apoptosis induction. We found that the adenylation level of *miR-7-5p* was significantly increased in cells overexpressing wild-type PAPD5, but not in cells overexpressing the catalytic-inactive mutant (Fig. 3e). Moreover, protein level of TAB2 (the target of *miR-7-5p*) was found to be elevated only in the wild-type, but not mutant, PAPD5-overexpressing cells (Fig. 3c, 3d). Taken together, our results show that PAPD5-induced apoptosis is dependent on its adenylation activity.

4. Fig 2A. How were miRNA levels measured and how is this list of 184 miRNAs determined? Are these highly expressed miRNAs in this cell type? In addition to heatmap showing fold-changes, miRNA expression levels should be provided.

Response: We thank the reviewer for his/her comments. We have performed a miRNA qPCR array (miRCURY LNA miRNA miRNome PCR Panels; YAHS-312YE-8; Qiagen) to determine the miRNA levels. Out of the 751 human miRNAs analyzed, the levels of 186 miRNAs were found to be downregulated in mutant CAG RNA-expressing cells, and their levels of reduction could be reversed upon *PAPD5* knockdown. The expression levels of these 186 miRNAs are now summarized in Supplementary Data 1. The raw cycle threshold values have also been deposited to GEO under the accession number of GSE271666.

5. To examine whether PAPD5 directly targets these downregulated miRNAs, authors should perform NGS and measure the changes on 3' adenylation of these miRNAs.

Response: We totally agree with the reviewer's viewpoint. Small RNA-seq has been performed to measure the 3' adenylation levels of the miRNAs in cells overexpressed with wild-type PAPD5 protein. We found that, among the 186 miRNAs which showed reduction in levels in mutant CAG RNA-expressing cells, the adenylation level of 11 of them, including *miR-7-5p* which targets TAB2, was significantly upregulated in wild-type PAPD5-expressing cells (Fig. 3e and Supplementary Fig. 5). Notably, such upregulation was not observed in cells overexpressed with the PAPD5 catalytic-inactive mutant (Fig. 3e and Supplementary Fig.

5). This suggests that the levels of these 11 miRNAs are directly regulated via the PAPD5-mediated adenylation mechanism. The expression of the remaining miRNAs that are downregulated in mutant CAG RNA cells and recovered upon *PAPD5* knockdown would be regulated via other mechanisms.

6. Fig 5E. Authors should include the mutant promoter luc-reporter here.

Response: We appreciate the reviewer's insightful comments. A *PAPD5* promoter luciferase reporter with YY1-binding site mutated is now included in our study as a control. We found that mutant CAG RNA failed to induce the promoter activity of this mutant luciferase reporter construct (Fig. 6h). This finding further highlights the critical role of YY1 in the mutant CAG RNA-mediated *PAPD5* upregulation.

7. Line338. I am not aware that PAPD5 is implicated in TDMD. Please provide refs.

Response: We apologize for the misleading information. This description has been removed from the revised manuscript, and the "Discussion" section has been revised (Lines 345 to 349).

Minor points:

1. According to HUGO-approved nomenclature, PAPD5 should be referenced as TENT4B. This should be at least mentioned in introduction.

Response: We thank the reviewer for pointing this out. This information is now included in the "Introduction" section (Lines 71 to 72) of the revised manuscript.

2. Figure and figure legends should provide essential information on how to interpret the results shown, rather than merely repeat the conclusions which are already stated in the main text. Using Fig1A as an example, it should be clearly indicated that the upper panel is a gel of RT-PCR and the bottom gel is a Western blot either in the figure itself or in the legend.

Response: We thank the reviewer for these remarks. The detailed interpretations are now included in figures and their respective figure legends of the revised manuscript.

Reviewer #3 (Remarks to the Authors):

1-None of the western blots (apart from Fig 6) are quantified. The standard for the field is to quantify replicates, display error bars, and describe differences with appropriate statistical analysis. This should be done for all western blots displayed.

Response: We greatly appreciate the reviewer for raising these remarks. We have now quantified all western blots displayed in the revised manuscript. The quantification results are included as Fig. 1c, 1e, 1k, 1o, 1r, 1t, 1w, 2b, 2e, 2h, 3d, 3i, 4h, 4k, 4n, 4p, 4s, 4v, 5c, 5e, 6d, 6j, 6l, 6n, 6p, 8a, 8d, 8k, 9d, 9f, 9i, 9l, 9o and 9r. The number of biological replicates, error bars, and the statistical analysis methods used are described in each figure legend.

2-Similarly for mRNA analysis throughout, this should be done by qPCR, not by showing a single, unquantified, sample

Response: We thank the reviewer for pointing this out. We have replaced all conventional RT-PCR results with RT-qPCR results. Specifically, the RT-qPCR results for the mRNA analysis throughout the study are now presented in Fig. 1a, 1g, 4a, 4d, 4f, 4i, 6e-g, 7a, 7d, 7g, 7j, 9g, 9j, 9m and 9p.

3-In all graphs displaying relative cell death the control samples can't be set to 1 as this eliminated any error bars from the control sample, thus prejudicing the analysis (an ANOVA can't be use if the samples have no error). It is fine to calculate the mean of the control sample, set that value to 1 but then all values need to be expressed relative to that average (including the control samples), so as to retain error bars for all samples.

Response: We appreciate the reviewer's insightful comments. The error bars are now included in the control samples of all the graphs displaying relative cell death (Fig. 1f, 1l, 1u, 1x, 2c, 2f, 3g, 4l, 4q, 4t, 4w, 5a, 5f, 8l).

4-Similarly for the quantification for YY1 in Fig 6, it is good that the western blots are quantified, but amounts either need to be expressed as quantified values, or normalised to the average of the control sample, as described above, but error bars have to be present for the control sample

Response: We thank the reviewer for his/her comments. The error bars are now included in both the control and experimental groups in all quantification analyses (Fig. 1c, 1e, 1k, 1o, 1r, 1t, 1w, 2b, 2e, 2h, 3d, 3i, 4h, 4k, 4n, 4p, 4s, 4v, 5c, 5e, 6d, 6j, 6l, 6n, 6p, 8a, 8d, 8k, 9d, 9f, 9i, 9l, 9o, 9r).

5- Can the number of rhabdomeres/ flies quantified please be annotated in each figure.

Response: We thank the reviewer for his/her comments. The number of rhabdomeres/flies quantified are now included in each figure (Fig. 1i, 4c, 5h, 7c, 7f, 7i, 7l, 7n).

6-Could the authors please comment as to why in Fig 5F and 5H, in the control situations where YY1 is down-regulated, there is no increase in PAPD5?

Response: We thank the reviewer for raising his/her concerns. We have now quantified the PAPD5 band intensities in Fig. 6i and 6m (original Fig. 5f and 5h), and found that in the control situations where *YY1* was knocked down, a ~3-fold induction in PAPD5 protein level was detected (Fig. 6j, 6n).

7-Can the authors please comment as to why in Fig 5R in the UAS-Httexon1 Q93 over-expressing flies, there is no decrease in YY1 transcript but in Fig 6A the same flies show a reduced YY1 expression. Does that mean YY1 is regulated at the level of translation?

Response: We thank the reviewer for his/her comments. We found that the soluble YY1 protein level was reduced in the HD fly model (Fig. 8a), while the *YY1* transcript level remained unchanged (Fig. 7d, 7j). Our data highlight that YY1 transcription is not perturbed in the mutant flies. The decrease in soluble YY1 protein level would be due to the recruitment of YY1 to the polyQ protein aggregates in HD models (Fig. 8c, 8d, 8g).

8-In Fig 5G and 5I it would be good to see a western blot with a YY1 antibody (rather than myc), to show the effect of the mutations on the endogenous YY1 and how much is YY1 increased with the over-expression of YY1-myc.

Response: We totally agree with the reviewer on this point. We have used an anti-YY1 antibody (ab109237; Abcam) to detect the YY1 protein level in Fig. 6k and 6o (original Fig. 5g and 5i). Quantification of the YY1 band intensity demonstrates a ~3-fold reduction of endogenous YY1 protein level in EGFP-Htt1-550Q89(CAG)-expressing cells (Fig. 6p). The endogenous YY1 level remained unchanged in mutant CAG RNA cells (Fig. 6l). We also show a ~7-fold induction of YY1 protein level in cells overexpressed with YY1-myc (Fig. 6l, 6p).

9-In Fig 5H there should a drop of YY1 in the EGFP-Htt1-550 Q89(CAG) condition, why is it not? The authors should comment

Response: We thank the reviewer for pointing this out. We have quantified the YY1 band intensities of all the experimental replicates, and a significant reduction (~3-fold) of endogenous YY1 protein level was observed in EGFP-Htt1-550Q89(CAG)-expressing cells (Fig. 6n). We have included a representative blot in the revised Fig. 6m (original Fig. 5h).

10- The co-localisation in Fig 6E and 6G should be analysed for statistical significance, for example with a Pearson correlation coefficient

Response: We totally agree with the reviewer on this point. We have performed Pearson's correlation coefficient analysis to investigate the co-localization between the endogenous YY1 protein and CAG RNA foci in Fig. 8e (original Fig. 6e and 6g). The analysis shows a positive correlation between YY1 protein expression and CAG RNA foci intensity ($P < 0.0001$), which supports our conclusion that there is a co-localization between the YY1 protein and CAG RNA foci (Fig. 8f).

11-To formally prove that PAPD5 and YY1 are in a linear pathway and not contributing to the disease formation separately from each other, it would good to show that RNAi of PAPD5, together with over-expression of YY1 does not lead to additive effects, the two interventions should be epistatic to each other. This experiment could easily be done in the fly eye model

Response: We thank the reviewer for his/her comments. Our results showed that the simultaneous overexpression of *dYY1* and knockdown of *dPAPD5* in the HD fly model did not show any additive suppression effect when compared with the single gene manipulation conditions (Fig. 7m, 7n). Our finding supports that YY1 and PAPD5 function in a linear pathway in HD pathogenesis.

12-The authors carry out a comprehensive analysis using a fly eye as disease model, however this is very much a developmental phenotype, whereas HD is an adult-onset disease. The disease relevance of the paper would be much greater if the authors showed the effects of PAPD5 down-regulation on an adult phenotype for the fly HD model (lifespan/climbing/activity). This is not essential.

Response: We thank the reviewer for his/her comments. In the revised manuscript, we employed the GeneSwitch system to induce the expression of the *HttQ93* transgene in the adult stage. We then evaluated the impact of *dPAPD5* knockdown (Fig. 4d) using the adult climbing assay. Our data demonstrate that the climbing defects of HD flies could be suppressed upon *dPAPD5* knockdown (Fig. 4e). These findings confirm that *dPAPD5* downregulation ameliorates the adult degenerative phenotype of the fly HD model.

POINT-BY-POINT RESPONSE TO REVIEWERS' COMMENTS

Reviewer #1 (Remarks to the Authors):

1. To show that PAPD5 is required for the miR-7 adenylation, they need to show PAPD5 loss-of-function effects on miRNA adenylation and abundance, rather than overexpression. The author should also show the global analysis of their miRNA array and sequencing data (e.g., volcano plot) and point out the miRNAs of interest.

Response: We greatly appreciate the reviewer for pointing out these issues. In the revised manuscript, we demonstrate that the adenylation of *miR-7-5p* is reduced upon knockdown of *PAPD5*, confirming its essential role in regulating *miR-7-5p* adenylation (Fig. 3i). Additionally, to provide a comprehensive analysis of our array and sequencing data, we have included: 1) a heat map analysis of miRNA array data (Fig. 3a and Supplementary Fig. 2), and 2) scatter plots of small RNA-seq data (Fig. 3g, 3h). The miRNA of interest, *miR-7-5p*, is highlighted in both the heat map analysis (Fig. 3a) and the scatter plots (Fig. 3g, 3h).

2. The ChIP-seq quantification says "read counts". These read counts should be normalized to sequencing depth. In addition, they should show some form of global analysis and point out their genes of interest.

Response: We thank the reviewer for pointing these out. The "read counts" in our ChIP-seq quantification have been normalized to the sequencing depth. We clarified this point in the methods section (Lines 663 to 664) of the revised manuscript, and modified the description in the quantification analysis to "Normalized read counts" (Fig. 8j). Additionally, we have included a scatter plot to illustrate the global changes in YY1's binding to gene promoters in the HD cell model, based on our ChIP-seq data (Fig. 8h). Our gene of interest, *PAPD5*, is highlighted in the scatter plot (Fig. 8h).

Reviewer #2 (Remarks to the Authors):

However, my primary concern regarding the direct involvement of PAPD5 in the downregulation of miRNAs in Huntington's Disease remains unresolved. The authors have indeed shown that only a subset of the downregulated miRNAs (if not only miR-7) is directly adenylated by PAPD5, which aligns with my original assertion that the widespread miRNA downregulation is more likely an indirect consequence of cellular toxicity rather than a direct effect of PAPD5 upregulation. Hence, the manuscript still lacks clarity on how PAPD5 contributes to the pathology of HD, a key point of this study. Therefore, I am unable to recommend this manuscript for publication in its current form.

Response: We greatly appreciate the reviewer's insightful comments. Our findings demonstrate that the level of PAPD5 is elevated in the HD iPSCs-derived striatal neurons (Fig. 9c, 9d) and HD patient striatal tissues (Fig. 9h, 9i). Notably, knockdown of *PAPD5* suppresses caspase-3 activation in HD patient striatal neurons (Fig. 9e, 9f) and alleviates retinal degeneration (Fig. 4b, 4c) and motor dysfunction (Fig. 4e) in our HD fly model. These results underscore the role of PAPD5 in modulating HD neurodegeneration.

Mechanistically, PAPD5 regulates the adenylation and expression of 11 miRNAs, including *miR-7-5p* (Fig. 3f and Supplementary Fig. 6). We showed that overexpression of wild-type PAPD5, but not its catalytic-inactive mutant, promotes the adenylation of *miR-7-5p* (Fig. 3f), while knockdown of *PAPD5* exerts the opposite effect (Fig. 3i). In our HD cell model, the level of *miR-7-5p* is reduced, but it is restored upon *PAPD5* knockdown (Fig. 3a). Furthermore, overexpression of *miR-7* mitigates cell death by counteracting the PAPD5-mediated apoptotic pathway in HD cells (Fig. 4u, 4v). Collectively, our results support the notion that PAPD5-induced dysregulation of *miR-7-5p* contributes to apoptotic induction in HD (Fig. 10).

In our study, we demonstrated that the widespread downregulation of miRNAs could be reversed upon *PAPD5* knockdown in our HD cell model (Fig. 3a). In addition to being directly regulated by the PAPD5-catalyzed adenylation, the downregulation of miRNAs may also be mediated through other pathways which involve PAPD5 in a less direct way. We found that multiple PAPD5-adenylated miRNAs (Supplementary

Fig. 6) modulate the expression of chromatin modifiers, which subsequently influence the expression of other miRNAs (1-9). For example, *miR-589-5p* and *miR-101-3p* negatively regulate the expression of *histone deacetylase 3 (HDAC3)* (9) and *histone deacetylase 9 (HDAC9)* (3), respectively. Both HDAC3 and HDAC9 are known to suppress the expression of various miRNAs, including members of the miR-17~92 cluster, such as *miR-17-5p*, *miR-18a-5p*, *miR-19b-1-5p*, and *miR-20a-5p* (10, 11). Therefore, PAPD5 induction leads to the adenylation and degradation of *miR-589-5p* and *miR-101-3p*, which further results in the increased levels of HDAC3 and HDAC9 followed by the degradation of the miR-17~92 cluster members. Indeed, these miR-17~92 cluster members were found to be downregulated in our HD cell model and their expressions were restored following *PAPD5* knockdown (Fig. 3a). Taken together, the broad downregulation of miRNAs can be triggered by both the direct adenylation effect of PAPD5, and the secondary pathways indirectly influenced by PAPD5. These mechanisms are now discussed in the revised manuscript between Lines 365 to 377.

References:

1. Fu X, Mao X, Wang Y, Ding X, Li Y. Let-7c-5p inhibits cell proliferation and induces cell apoptosis by targeting ERCC6 in breast cancer. *Oncol Rep.* 2017;38(3):1851-6.
2. Wang Y, Zhang G, Li P, Kang L, Qin B, Cao Y, et al. Profiling and Integrated Analysis of the ERCC6-regulated circRNA-miRNA-mRNA Network in Lens Epithelial Cells. *Curr Eye Res.* 2021;46(9):1341-52.
3. Jin Q, He W, Chen L, Yang Y, Shi K, You Z. MicroRNA-101-3p inhibits proliferation in retinoblastoma cells by targeting EZH2 and HDAC9. *Exp Ther Med.* 2018;16(3):1663-70.
4. Liu T, Cai J, Cai J, Wang Z, Cai L. EZH2-miRNA Positive Feedback Promotes Tumor Growth in Ovarian Cancer. *Front Oncol.* 2020;10:608393.
5. Wang J, Wang S, Zhou J, Qian Q. miR-424-5p regulates cell proliferation, migration and invasion by targeting doublecortin-like kinase 1 in basal-like breast cancer. *Biomed Pharmacother.* 2018;102:147-52.
6. Sureban SM, May R, Qu D, Weygant N, Chandrakesan P, Ali N, et al. DCLK1 regulates pluripotency and angiogenic factors via microRNA-dependent mechanisms in pancreatic cancer. *PLoS One.* 2013;8(9):e73940.
7. Zhao Y, Zhu C, Chang Q, Peng P, Yang J, Liu C, et al. MiR-424-5p regulates cell cycle and inhibits proliferation of hepatocellular carcinoma cells by targeting E2F7. *PLoS One.* 2020;15(11):e0242179.
8. Mitxelena J, Apraiz A, Vallejo-Rodriguez J, Malumbres M, Zubiaga AM. E2F7 regulates transcription and maturation of multiple microRNAs to restrain cell proliferation. *Nucleic Acids Res.* 2016;44(12):5557-70.
9. Rahbari R, Rahimi K, Rasmi Y, Khadem-Ansari MH, Abdi M. miR-589-5p Inhibits Cell Proliferation by Targeting Histone Deacetylase 3 in Triple Negative Breast Cancer. *Arch Med Res.* 2022;53(5):483-91.
10. Wang Y, Frank DB, Morley MP, Zhou S, Wang X, Lu MM, et al. HDAC3-Dependent Epigenetic Pathway Controls Lung Alveolar Epithelial Cell Remodeling and Spreading via miR-17-92 and TGF-beta Signaling Regulation. *Dev Cell.* 2016;36(3):303-15.
11. Kaluza D, Kroll J, Gesierich S, Manavski Y, Boeckel JN, Doebele C, et al. Histone deacetylase 9 promotes angiogenesis by targeting the antiangiogenic microRNA-17-92 cluster in endothelial cells. *Arterioscler Thromb Vasc Biol.* 2013;33(3):533-43.

POINT-BY-POINT RESPONSE TO REVIEWERS' COMMENTS

Reviewer #1 (Remarks to the Authors):

The authors made an attempt to address my specific concerns. However, neither unbiased high-throughput measurements strongly supported their initial findings. The effect of PAPD5 KD on miR-7 abundance is so small (Fig.3g) that it could not possibly have any impact on target repression. A similar issue applies to their genome-wide YY1 ChIP-seq analysis, in which YY1 binding to PAPD5 promoter did not change substantially. This is not a request for additional analyses- the amount of analysis already makes the paper nearly impossible to read- but rather, I suggest the authors think carefully about their model and remove some of the components that their data cannot fully justify.

Response: We greatly appreciate the reviewer's insightful comments. We demonstrated that the level of *miR-7-5p* was reduced in our HD cell model, while such reduction was restored when *PAPD5* was knocked down (Fig. 3a). It has been reported that *PAPD5* decreases miRNA expression by promoting their 3' adenylation (1). We performed a small RNA-seq and found that the adenylation of *miR-7-5p* was induced by the wild-type *PAPD5* protein (Fig. 3f). Under this experimental condition, the protein level of TAB2, the target of *miR-7-5p*, was significantly elevated (Fig. 3d, e). Notably, neither *miR-7-5p* adenylation nor TAB2 protein induction was detected in cells expressing a catalytic-inactive *PAPD5* mutant (Fig. 3d-f). These findings strongly suggest that *PAPD5* facilitates the adenylation of *miR-7-5p*, leading to its reduction, which in turn triggers the induction of TAB2. Furthermore, we observed a reduction in *miR-7-5p* (Fig. 9j) and an induction of TAB2 (Fig. 9k, l) in striatal samples from HD patients, where *PAPD5* level was significantly increased (Fig. 9g-i). Our data support our working model that *PAPD5* mediates miRNA dysregulation in HD pathogenesis.

Our data further demonstrate that *PAPD5* directly promotes the adenylation of another ten miRNAs (Supplementary Fig. 6) and causes their reduction in levels in our HD cell model (Fig. 3a). Notably, the adenylation of all these *PAPD5*-targeted miRNAs did not change substantially (Fig. 3f and Supplementary Fig. 6). This can be attributed to the rapid degradation of adenylated miRNAs. We have included a discussion of this point in the revised manuscript between lines 365 to 367.

We investigated the underlying mechanisms of *PAPD5* dysregulation and showed that YY1, a *PAPD5* transcriptional repressor (Fig. 6b-e), was recruited to both CAG RNA foci and Htt-polyQ protein aggregates in HD striatal neurons (Fig. 9a, b). Additionally, we demonstrated a reduced binding of YY1 to the *PAPD5* gene promoter under the disease condition (Fig. 8h, i), leading to the induction of *PAPD5* (Fig. 8c, d) and activation of the *PAPD5*-mediated apoptotic pathway (Fig. 8j, k). More importantly, overexpression of YY1 suppressed *PAPD5* upregulation and mitigated apoptotic induction in our HD cell model (Fig. 6o, p). These findings underscore the impairment of the YY1-*PAPD5* regulatory axis in contributing to neuronal apoptosis in HD.

YY1 recruits a series of epigenetic modifiers to gene promoters to inhibit transcription (2, 3). The depletion of YY1's binding to promoter regions markedly reduces the enrichment of these epigenetic modifiers, thereby enhancing gene transcriptional activation (4). Although the binding of YY1 to the *PAPD5* promoter did not change substantially in the disease cells (Fig. 8i), it is sufficient to trigger *PAPD5* induction (Fig. 8c, d), presumably by alleviating the epigenetic repression at the *PAPD5* promoter. We have also included a discussion of this point in the revised manuscript between lines 394 to 400.

References

1. Boele J, Persson H, Shin JW, Ishizu Y, Newie IS, Sokilde R, et al. *PAPD5*-mediated 3' adenylation and subsequent degradation of miR-21 is disrupted in proliferative disease. *Proc Natl Acad Sci U S A*. 2014;111(31):11467-72.
2. Phongbunchoo Y, Braikia FZ, Pessoa-Rodrigues C, Ramamoorthy S, Ramachandran H, Grosschedl A, et al. YY1-mediated enhancer-promoter communication in the immunoglobulin mu locus is regulated by MSL/MOF recruitment. *Cell Rep*. 2024;43(7):114456.
3. Schlesinger S, Lee AH, Wang GZ, Green L, Goff SP. Proviral silencing in embryonic cells is regulated by Yin Yang 1. *Cell Rep*. 2013;4(1):50-8.

4. Dong X, Guo R, Ji T, Zhang J, Xu J, Li Y, et al. YY1 safeguard multidimensional epigenetic landscape associated with extended pluripotency. *Nucleic Acids Res.* 2022;50(21):12019-38.